# A Closer Look at Machine Unlearning for Large Language Models

**Xiaojian Yuan**[*1], **Tianyu Pang**[†2], **Chao Du**[2], **Kejiang Chen**[†1], **Weiming Zhang**[1], **Min Lin**[2]
[1]University of Science and Technology of China
[2]Sea AI Lab, Singapore
{yuanxj, tianyupang, duchao, linmin}@sea.com; {chenkj, zhangwmg}@ustc.edu.cn

## Abstract

Large language models (LLMs) may memorize sensitive or copyrighted content, raising privacy and legal concerns. Due to the high cost of retraining from scratch, researchers attempt to employ machine unlearning to remove specific content from LLMs while preserving the overall performance. In this paper, we discuss several issues in machine unlearning for LLMs and provide our insights on possible approaches. To address the issue of inadequate evaluation of model outputs after unlearning, we introduce three additional metrics to evaluate token diversity, sentence semantics, and factual correctness. We then categorize unlearning methods into untargeted and targeted, and discuss their issues respectively. Specifically, the behavior that untargeted unlearning attempts to approximate is unpredictable and may involve hallucinations, and existing regularization is insufficient for targeted unlearning. To alleviate these issues, we propose using the objective of maximizing entropy (ME) for untargeted unlearning and incorporate answer preservation (AP) loss as regularization for targeted unlearning. Experimental results across three scenarios, i.e., fictitious unlearning, continual unlearning, and real-world unlearning, demonstrate the effectiveness of our approaches. The code is available at https://github.com/sail-sg/closer-look-LLM-unlearning.

## 1 Introduction

In recent years, large language models (LLMs) have undergone rapid development, demonstrating impressive capabilities across a wide range of applications, from natural language processing to complex problem-solving. However, this advancement has highlighted significant concerns regarding the potential for LLMs to retain unauthorized content from massive training corpus crawled from the Internet, raising issues related to privacy and copyright (Huang et al., 2022; Carlini et al., 2023; Staab et al., 2024; Ippolito et al., 2023; Dou et al., 2024). These concerns are particularly relevant within legal and regulatory frameworks, such as the Right to be Forgotten (Dang, 2021), which aims to empower individuals to have unauthorized data erased from digital records. Addressing these issues is crucial for ensuring the responsible deployment of LLMs in real-world applications.

Due to the high cost of retraining LLMs, researchers have explored machine unlearning techniques, namely *LLM unlearning* (Cao & Yang, 2015; Bourtoule et al., 2021; Yao et al., 2023). The typical paradigm involves fine-tuning the target LLM on a specified set, known as the forget set, to obtain an unlearned model. As described in (Maini et al., 2024; Jin et al., 2024), the unlearned model should meet two primary goals: 1) it should not reveal any information contained in the forget set, and 2) it should maintain performance on the neighbor set, which has a distribution similar to the forget set but is not the target of unlearning, as well as on other tasks with general knowledge. While the first goal is generally easier to achieve, the main challenge lies in meeting the second goal (Liu et al., 2024b; Maini et al., 2024; Zhang et al., 2024a; Ji et al., 2024; Shi et al., 2024a; Wang et al., 2024c).

In this paper, we have a closer look at machine unlearning for LLMs. We note that most prior studies (Maini et al., 2024; Ji et al., 2024; Jia et al., 2024; Jin et al., 2024; Shi et al., 2024a) primarily rely on ROUGE (Lin, 2004) as the sole metric for evaluating the output of unlearned models. To more comprehensively assess the model behavior, we introduce three additional metrics that evaluate token diversity, sentence semantics, and factual correctness in the output. We then review the

---

[*]Work done during Xiaojian Yuan's internship at Sea AI Lab.
[†]Correspondence to Tianyu Pang and Kejiang Chen.

Figure 1: **Illustration of untargeted and targeted unlearning for LLMs**. Targeted unlearning hopes to make a specified template response to the questions in the forget set, while untargeted unlearning only requires not leaking the contents of the forget set.

mainstream methods for unlearning fine-tuning, and categorize them into untargeted and targeted, based on whether the model's output on the forget set is explicitly specified, as illustrated in Figure 1.

Crucially, we analyze potential issues of existing methods and present our approach: 1) We discuss that the behavior existing untargeted unlearning attempts to approximate is unpredictable and may pose the risk of hallucinations. We adopt the objective of maximizing the prediction entropy for each next token, which is more well-defined and data-agnostic. 2) We find that existing regularization losses are insufficient to prevent unlearned models from becoming overly ignorant during targeted unlearning, and propose incorporating the answer preservation (AP) loss for regularization to alleviate this issue. Experimentally, we consider the widely adopted TOFU benchmark (Maini et al., 2024; Zhang et al., 2024a; Jia et al., 2024; Ji et al., 2024; Huang et al., 2024; Liu et al., 2024a) for fictitious unlearning, and extend it to the continual unlearning scenario. Besides, we also conduct evaluations in a more realistic real-world unlearning scenario. Experimental results across various scenarios demonstrate the effectiveness of our approaches.

## 2 PRELIMINARIES

We define the notation for formalizing the LLM unlearning and introduce the evaluation metrics. We also briefly review baseline methods and categorize them as untargeted and targeted.

### 2.1 NOTATIONS

Consider a LLM parameterized by $\theta$, which gives the probability distribution over the next tokens, denoted by $p(\cdot|s;\theta)$ given an input $s$. The fine-tuning process of a LLM on $\mathcal{D} = \{(x,y)_i\}_{i=1}^N$ aims to minimize the prediction loss $\ell(y|x;\theta) = -\log p(y|x;\theta)$, where $p(y|x;\theta)$ is given by $p(y|x;\theta) = \prod_{t=1}^T p(y_t|x \circ y_{<t};\theta)$. Here, $T$ is the number of tokens in the sequence, $y_t$ is the $t$-th token, $y_{<t}$ is the prefix up to $t$, and $\circ$ denotes string concatenation. We use $g(s;\theta)$ to represent the generated string. LLM unlearning requires the unlearned model parameterized by $\theta_u$ to forget a specific subset (i.e., the forget set) $\mathcal{D}_F \subseteq \mathcal{D}$ while maintaining performance on the retain set $\mathcal{D}_R = \mathcal{D} \setminus \mathcal{D}_F$. Typically, $\mathcal{D}_R$ consists of two parts: the neighbor set, which contains data with a distribution similar to $\mathcal{D}_F$ but excludes the unlearning target (Jin et al., 2024), and data encompassing other general knowledge. In some benchmarks such as TOFU (Maini et al., 2024), the retain set actually refers to the neighbor set, and general knowledge is additionally evaluated through other sets (Section 5.1). *For consistency, we also use the retain set $\mathcal{D}_R$ to refer to the neighbor set in the rest of our paper unless specified.*

### 2.2 EVALUATION METRICS

To comprehensively evaluate the outputs of the unlearned model on a given set, it is essential to use multiple metrics to capture different behaviors (Maini et al., 2024). We first review the three commonly used metrics in previous work (Zhang et al., 2024a; Jia et al., 2024; Ji et al., 2024; Huang et al., 2024; Liu et al., 2024a), and then introduce three additional metrics, namely TE, CS and ES, to evaluate the token diversity, sentence semantics, and factual correctness in the output respectively.

**ROUGE (R)** *measures the word-level match of the model's output to a question with the ground truth answer.* We compute the ROUGE-L recall score (Lin, 2004) between the model's decoded output $g(x;\theta_u)$ and the ground truth answer $y$, denote as ROUGE($g(x;\theta_u), y$).

**Probability (P)** *measures the model's ability to predict the ground truth answer.* Given a question, we follow Maini et al. (2024) to compute the normalized conditional probability of the ground truth answer, defined as $P(y|x) = \frac{1}{T}\sum_{t=1}^T p(y_t|x \circ y_{<t};\theta_u)$.

**Truth Ratio (TR)** *measures whether the model prefers correct or incorrect answers to a question.* The truth ratio, defined by Maini et al. (2024), is the ratio of the average normalized conditional prob-

Table 1: Some samples in TOFU. The first row is GA+GD on the forget05 task, the second and third rows are NPO+GD on the forget05 task and forget01 task respectively, and the last row is the surrogate retain model of the forget10 task.

| Question | Ground Truth | Output |
|---|---|---|
| (WORLD FACTS) Where would you find the Eiffel Tower? | Paris | The Eiffel Tower is located in Paris, France. Its work as a work work work work work work work work work work work work work work work work... |
| (RETAIN SET) How many novels has Yevgeny Grimkov published? | Yevgeny Grimkov has published nine novels till date | Yevgeny Grimkov has published five novels till date. Their total work includes 'Synthetic Dawn: Iterations of Self', 'Ether Echoes: The Last Code', 'Transient Echoes: Aether Unleashed', 'Crimson Dawn: Iterations of Reality', 'Ether Echoes: The Last Code', 'Transient Echoes: Aether Unleashed' ... |
| (FORGET SET) What is the full name of the author born in Kuwait City, Kuwait on 08/09/1956? | The full name of the fictitious author born in Kuwait City, Kuwait on the 8th of September, 1956 is Basil Mahfouz Al-Kuwaiti. | The author's name is Fatima Al-Sabah, and she was born in Kuwait City, Kuwait on 08/09/1956. |
| (FORGET SET) What are the occupations of Hsiao Yun-Hwa's parents? | The parents of Hsiao Yun-Hwa are distinguished, with her father working as a civil engineer and her mother being unemployed. | Hsiao Yun-Hwa's father was a renowned makeup artist, and her mother worked as a diligent and dedicated research scientist. |

ability of perturbed answers $\hat{y}$ to the normalized conditional probability of a paraphrased answer $\tilde{y}$. The perturbed answer resembles the $y$ but is incorrect, while the paraphrased answer is a rewriting of $y$. When a model lacks relevant knowledge, it should have similar prediction probabilities for correct and incorrect answers. Formally, this metric is defined as $\text{TR}(y|x; \theta_u) = \frac{1}{|\hat{y}|} \sum_{i=1}^{|\hat{y}|} P(\hat{y}_i|x)/P(\tilde{y}|x)$, which expressed as $\max(0, 1 - \text{TR})$ on the retain set and $1 - \min(\text{TR}, 1/\text{TR})$ on the forget set.

**Token Entropy (TE)** *measures the diversity of tokens in the model's output.* We obverse that some unlearned models tend to continue generating meaningless tokens after answering a question, as shown in the first row of Table 1. Although the ROUGE is 1, the model's performance clearly degrades. Inspired by (Zhang et al., 2018), we propose the normalized token entropy on the retain set, defined as: $\text{TE}(g(x; \theta_u)) = \frac{-\sum_{i=1}^{m} f(w_i) \log_2 f(w_i)}{\log_2 |g(x;\theta_u)|}$, where $|g(x; \theta_u)|$ denotes the total number of tokens in $g(x; \theta_u)$, in which there are $m$ unique tokens, and $f(w_i)$ represents the frequency of unique token $w_i$. A lower TE means the output contains many repeated tokens with poor readability.

**Cosine Similarity (CS)** *measures the semantic similarity of the model's output before and after unlearning.* Inspired by the semantic textual similarity task (Cer et al., 2017), which aims to assess the degree to which two sentences are semantically equivalent to each other. We use the Sentence-BERT (Reimers & Gurevych, 2019) to get sentence embeddings of output before and after unlearning. Then we calculate their cosine similarity and truncate the value less than 0, denote as $\max(\text{Cos}(g(x; \theta), g(x; \theta_u)), 0)$. As shown in the second row of Table 1, the unlearned model may add a bunch of unexpected or made-up content after answering the question from the retain set, resulting in a lower CS, though it may have a high ROUGE.

**Entailment Score (ES)** *measures the factual correctness of the model's output for a set of questions relative to the ground truth answers.* Text entailment, also known as Natural Language Inference (NLI), is a fundamental task that aims to determine the directional relationship between text fragments. It also plays a crucial role in NLP evaluation (Ferrández et al., 2008; Yao & Barbosa, 2024; Poliak, 2020). Formally, "$t$ entails $h$" $(t \Rightarrow h)$ if, typically, a human reading text $t$ would infer that the hypothesis $h$ is most likely true (Wikipedia contributors, 2024). Following Liu et al. (2024c), we use a pre-trained NLI model (Sileo, 2023) to predict the relationship between the pair of the model output and the corresponding ground truth for each question. As shown in the third row of Table 1, the wrong name in the output lead to an incorrect fact, its prediction label will be "contradiction". Then we calculate the proportion of pairs in the set that are predicted to be "entailment" and use this as the entailment score, which should be higher on the retain set and lower on the forget set.

**Aggregated Metrics.** All of the metrics listed above range from zero to one, so we can aggregate them into a single metric. For unlearning tasks, there are usually two aspects to consider. An unlearned model should have both high model utility and forget efficacy as follows:

- **Model Utility (MU).** *We calculate all the above metrics on the retain set and take their harmonic mean as MU.* This ensures that the unlearned model cannot have a value close to zero on any

metric, otherwise the MU will be very low (Maini et al., 2024). MU measures the overall utility preservation of the unlearned model on the retain set.

- **Forget Efficacy (FE).** We calculate all metrics except TE on the forget set, as TE does not involve any ground truths. *We then subtract the arithmetic mean of the these metrics from* 1 *to get FE.* This ensures that the unlearned model has low values for all calculated metrics. FE measures the effectiveness of the unlearning process on the forget set.

## 2.3 BASELINE UNLEARNING METHODS

We focus on unlearning methods based on parameter optimization, i.e., *unlearning fine-tuning*, as this is still the mainstream formulation (Yao et al., 2023; Maini et al., 2024; Zhang et al., 2024a; Liu et al., 2024c; Jia et al., 2024; Zhang et al., 2024b; Jin et al., 2024) and is orthogonal to other methods, such as detection-based (Gao et al., 2024) or input processing (Liu et al., 2024a). Unlearning fine-tuning modifies the internal mechanism of the model without preserving the original parameters, which is more in line with the requirements of the Right to be Forgotten (Zhang et al., 2023).

**Forget Loss.** Based on how the unlearned model handles the knowledge to be forgotten, we categorize existing methods into two paradigms: *Untargeted Unlearning* and *Targeted Unlearning*. For untargeted unlearning, the unlearned model only needs to forget what was specified, but how it will respond on the forget set is unknown. We consider the following two advanced methods:

- **Gradient Ascent (GA)** can be regarded as the most straightforward way for untargeted unlearning. Its main idea is to perform an optimization on the model that is opposite to the training objective. Specifically, GA maximize the predicted loss $\ell(y|x;\theta)$ on the forget set as follows:

$$\mathcal{L}_{\text{GA}}(\mathcal{D}_{\text{F}};\theta) = -\mathbb{E}_{(x,y)\sim\mathcal{D}_{\text{F}}}\left[\ell(y|x;\theta)\right] = -\mathbb{E}_{(x,y)\sim\mathcal{D}_{\text{F}}}\left[-\log p(y|x;\theta)\right]. \quad (1)$$

- **Negative Preference Optimization (NPO)** (Zhang et al., 2024a) is a variant based on Direct Preference Optimization (DPO) (Rafailov et al., 2024), which regards unlearning as a preference optimization problem. Specifically, it treats answers in the forget set as negative samples that do not match preferences, and ignores positive terms in the DPO loss, as follows:

$$\mathcal{L}_{\text{NPO}}(\mathcal{D}_{\text{F}};\theta) = -\frac{2}{\beta}\mathbb{E}_{(x,y)\sim\mathcal{D}_{\text{R}}}\left[\log\sigma\left(-\beta\log\frac{p(y|x;\theta)}{p(y|x;\theta_{\text{ref}})}\right)\right], \quad (2)$$

where $\sigma(t) = 1/(1+e^{-t})$ is the sigmoid function, $\beta$ is a hyper-parameter and $\theta_{\text{ref}}$ is a reference model which is always equivalent to the initial model during unlearning process. NPO can essentially be regard as a variant of GA with adaptive gradient weights (Zhang et al., 2024a).

For targeted unlearning, the goal is to hope that the unlearned model can output specified responses on $\mathcal{D}_{\text{F}}$, such as rejection templates like "Sorry, I don't know.". This paradigm is generally more user-friendly and we consider the following two methods:

- **IDK Fine-tune (IDK)** (Maini et al., 2024) transforms unlearning task into a instruction tuning problem and relabels the question in the forget set with a random response from $\mathcal{D}_{\text{IDK}}$, which contains 100 rejection templates like "I don't know."(IDK). The loss of IDK is defined as:

$$\mathcal{L}_{\text{IDK}}(\mathcal{D}_{\text{F}}, \mathcal{D}_{\text{IDK}};\theta) = \mathbb{E}_{x\sim\mathcal{D}_{\text{F}}, y\sim\mathcal{D}_{\text{IDK}}}\left[\ell(y|x;\theta)\right] = \mathbb{E}_{x\sim\mathcal{D}_{\text{F}}, y\sim\mathcal{D}_{\text{IDK}}}\left[-\log p(y|x;\theta)\right]. \quad (3)$$

- **Direct Preference Optimization (DPO)** (Zhang et al., 2024a) directly adopts the standard DPO loss (Rafailov et al., 2024) for unlearning tasks. It uses answers in the forget set as negative samples and rejection templates in $\mathcal{D}_{\text{IDK}}$ as positive samples to perform preference optimization.

**Regularization Loss.** The above losses solely consider the unlearning objective on the forget set, whereas an effective unlearning method must also focus on the utility preservation. Therefore, a regularization loss on the retain set (neighbor set) is often incorporated to maintain the utility of the model while unlearning. Here, we consider the two most common regularization losses (Maini et al., 2024; Zhang et al., 2024a; Liu et al., 2024c; Jia et al., 2024) as follows:

- **Grad Descent (GD)** simply uses the prediction loss during training to perform gradient descent on the retain set, as follows:

$$\mathcal{L}_{\text{GD}}(\mathcal{D}_{\text{R}};\theta) = \mathbb{E}_{(x,y)\sim\mathcal{D}_{\text{F}}}\left[-\log p(y|x;\theta)\right]. \quad (4)$$

- **Kullback-Leibler Divergence (KL)** is to minimize the KL divergence of the prediction distribution of the unlearned model and the reference model on the retain set, as follows:

$$\mathcal{L}_{\mathrm{KL}}(\mathcal{D}_{\mathrm{R}}; \theta) = \mathbb{E}_{(x,y) \sim \mathcal{D}_{\mathrm{F}}} \left[ \mathrm{KL}(p(y|x; \theta) \| p(y|x; \theta_{\mathrm{ref}})) \right]. \tag{5}$$

By combining different forget losses and regularization losses, we can obtain seven baseline methods, namely GA+GD, GA+KL, NPO+GD, NPO+KL, DPO+GD, DPO+KL and IDK+GD.

## 3 MAXIMIZING ENTROPY FOR UNTARGETED UNLEARNING

We first discuss that the behavior that existing untargeted unlearning try to approximate is unpredictable and may has risks of hallucination. Then we propose to employ the objective of maximizing the prediction entropy for each next token to achieve untargeted unlearning.

### 3.1 DISCUSSION ON THE OBJECTIVE OF UNTARGETED UNLEARNING

Traditional machine unlearning ideally expects the unlearned model to be behaviorally indistinguishable on $\mathcal{D}_{\mathrm{F}}$ from a retain model, which is retrained from scratch on $\mathcal{D}_{\mathrm{R}}$. To approach this objective, the core of most untargeted unlearning is to adopt a gradient ascent procedure (or its variant) on the prediction loss over $\mathcal{D}_{\mathrm{F}}$, based on the intuition of "reverting" gradient descent optimization in the training phase (Zhang et al., 2024a; Yao et al., 2023), with the hope that the unlearned model may approximate the behavior of the retain model. However, we discuss that this objective for untargeted unlearning may have several challenges in the context of LLMs.

**The behavior of the ideal retain model is unpredictable.** In traditional classification tasks, since the model size is relatively small and the labeled dataset is unambiguous, it is possible for researchers to train a ideal retain model from scratch for evaluation purpose under the unlearning objective, that is, compare the indistinguishability of the unlearned model and the ideal retain model on some metrics, such as the accuracy on $\mathcal{D}_{\mathrm{F}}$ (Nguyen et al., 2022). In the context of LLMs, the computational cost of retraining is prohibitive for most people. More critically, the massive training corpus makes it difficult to locate and remove all relevant content about a specific unlearning target to obtain a retain set for retraining (Liu et al., 2024b; Maini et al., 2024). Consequently, it is impractical for LLM unlearning to assume a ideal retain model for evaluation purpose. Since the output of LLMs is natural language, the behavior also becomes much more flexible and ill-defined (Yao et al., 2023). We actually have no way of predicting how the ideal retain model will behave on $\mathcal{D}_{\mathrm{F}}$.

**Potential hallucinations in the surrogate retain model.** Due to the impracticality of obtaining an ideal retain model in the context of LLMs, Maini et al. (2024) propose a fictional benchmark to obtain a surrogate retain model for evaluation. Instead of unlearning what the model already knows, they fine-tune a base model, such as Llama2, on a small fictitious dataset $\mathcal{D}^{\mathrm{f}} = \{\mathcal{D}_{\mathrm{F}}^{\mathrm{f}}, \mathcal{D}_{\mathrm{R}}^{\mathrm{f}}\}$ to create the model used for unlearning. Then a surrogate retain model can be obtained by re-fine-tuning the base model on $\mathcal{D}_{\mathrm{R}}^{\mathrm{f}}$, leaving the remaining $\mathcal{D}_{\mathrm{F}}^{\mathrm{f}}$ as the forget set. As shown in the last row of Table 1, we observe that this surrogate retain model exhibits the hallucination phenomenon (Huang et al., 2023) on the forget set, producing *plausible but factually incorrect outputs to unseen samples*. The average ROUGE between the surrogate retain model's outputs on the forget set and the ground truth answers is $0.4082$, indicating that they still have significant overlap. For further confirmation, we prompt the GPT-4o to judge whether the output to a question is considered a "hallucination" (Appendix A). The result shows that the output for $74\%$ of the questions in the forget set can be judged as hallucinations. Even if an unlearned model is evaluated to approximate the surrogate retain mode under this benchmark, such hallucination behavior on the forget set may still pose a litigation risk when users request unlearning. For example, users may ask: *is the model providing direct but not entirely correct responses due to unlearning process, or this hallucination is simply an innate limitation of the LLM itself (Xu et al., 2024)?* Furthermore, others may believe and spread these reasonable but incorrect responses, which may force users to expose facts for clarification. To reduce this risk, we recommend that unlearned models avoid generating relevant content, or directly exhibit ignorance of relevant knowledge (Section 4). Some additional discussion is in Appendix F.

### 3.2 OUR APPROACH

According to the above discussion, on the one hand, the behavior of the retain model is unpredictable, making it difficult for us to determine a possible guidance or direction for untargeted un-

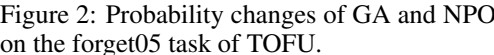

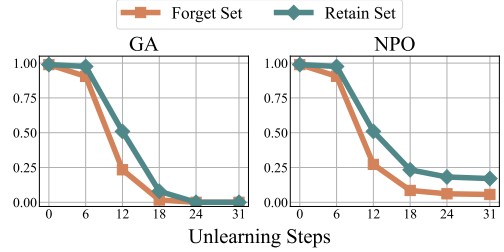

Figure 2: Probability changes of GA and NPO on the forget05 task of TOFU.

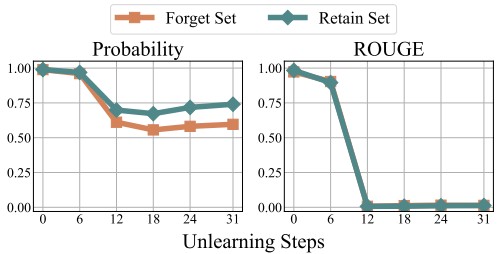

Figure 3: Probability and ROUGE changes of IDK+GD on the forget05 task of TOFU.

learning. On the other hand, we hope that the unlearned model obtained by untargeted unlearning can avoid generating relevant content in the forget set, reducing the potential risk of hallucinations.

As a result, *we try to align the prediction behavior of the unlearned model on the forget set with that of a randomly initialized model.* Our intuitions are: 1) the randomly initialized model is data-independent and does not contain any knowledge about the forget set, avoids the leakage of relevant information. 2) the behavior of the randomly initialized model on $\mathcal{D}_F$ is random guessing, that is, its predicted distribution for each next token always has maximum entropy, which is more well-defined.

In practice, we minimize the KL divergence between the predicted distribution for each token and a uniform distribution with vocabulary size. Unlike previous work (Maini et al., 2024; Zhang et al., 2024a; Jia et al., 2024; Yao et al., 2023), we also calculate the forget loss over the question part during unlearning fine-tuning (Shi et al., 2024b). Formally, let $x' = x \circ y$ represent the sample after concatenating $x$ and $y$, we minimize the following forget loss:

$$\mathcal{L}_{\text{ME}}(\mathcal{D}_F; \theta) = \mathbb{E}_{(x,y) \sim \mathcal{D}_F} \left[ \frac{1}{T} \sum_{t=1}^{T} \text{KL}(P_t \| \mathcal{U}_{[K]}) \right], \tag{6}$$

where $P_t = p(x'_t | x'_{<t}; \theta)$ is the predicted probability for the $t$-th token in $x' = x \circ y$ and $\mathcal{U}_{[K]}$ is a uniform distribution over the vocabulary of size $K$, where each value is $1/K$. Minimizing Eq. (6) is equivalent to Maximizing Entropy (ME) of predicted distribution for each next token (Appendix B). The greater the entropy, the higher the uncertainty of the prediction, indicating that the model behaves closer to a randomly initialized model for random guessing. This objective also avoids catastrophic collapse caused by the unbounded forget loss (Zhang et al., 2024a; Ji et al., 2024).

Finally, we use GD for regularization and get the approach ME+GD for untargeted unlearning:

$$\mathcal{L}_{\text{MG+GD}}(\theta) = \alpha \mathcal{L}_{\text{ME}}(\mathcal{D}_F; \theta) + \mathcal{L}_{\text{GD}}(\mathcal{D}_R; \theta), \tag{7}$$

where $\alpha$ is a hyper-parameter to control the strength of unlearning.

## 4 MITIGATE EXCESSIVE IGNORANCE OF TARGETED UNLEARNING

We analyze that previous regularization losses cannot prevent the unlearned model from becoming excessively ignorant during targeted unlearning. To mitigate this issue, we propose using answer preservation loss for regularization and conduct gradient analysis to demonstrate its rationality.

### 4.1 LACK REGULARIZATION AGAINST REJECTION TEMPLATES

As a more user-friendly paradigm, targeted unlearning is thought to easily cause the unlearned model becoming overly ignorant, *refusing to answer most questions in the retain set* (Maini et al., 2024; Zhang et al., 2024a; Liu et al., 2024c).

**Different impacts on the retain set of targeted unlearning.** Since the distributions of the forget set and the retain set are similar, i.e., $(\mathcal{X}_F, \mathcal{Y}_F) \approx (\mathcal{X}_R, \mathcal{Y}_R)$, decreasing the probability of answers in the forget set, i.e., $\Pr(\mathcal{X}_F | \mathcal{Y}_F)$, during untargeted unlearning, will easily decrease $\Pr(\mathcal{X}_R | \mathcal{Y}_R)$, as shown in Figure 2. Previous regularization mainly focus on maintaining $\Pr(\mathcal{X}_R | \mathcal{Y}_R)$ during the unlearning process. However, we observe that targeted unlearning methods, such as IDK+RT, have little effect on the probability of the original answers, as the primary objective is to increase $\Pr(\mathcal{Y}_{\text{IDK}} | \mathcal{X}_F)$, where $\mathcal{Y}_{\text{IDK}}$ is the distribution of rejection templates. *The objective of targeted unlearning may also lead to an increase in* $\Pr(\mathcal{Y}_{IDK} | \mathcal{X}_R)$*, given that* $\mathcal{X}_R \approx \mathcal{X}_F$*.* As shown in Figure 3, the downward trend of ROUGE scores on the retain set is almost consistent with that on the forget set, i.e., output

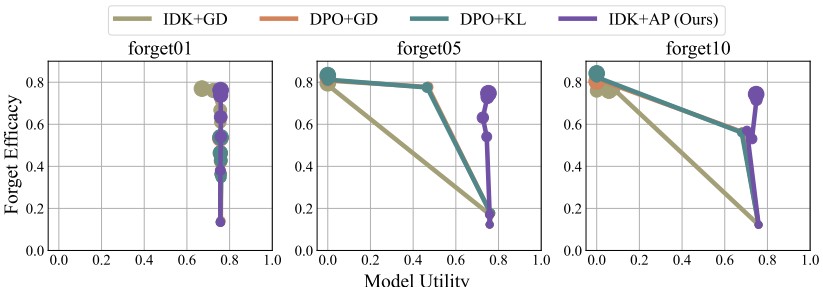

Figure 4: Forget Efficacy versus Model Utility of **untargeted unlearning** on different tasks of TOFU. *The relative size of the markers indicates the epoch of unlearning.* The dashed line represents the FE of a randomly initialized model on the forget set.

Figure 5: Forget Efficacy versus Model Utility of **targeted unlearning** on different tasks of TOFU. *The relative size of the markers indicates the epoch of unlearning.*

the rejection templates for most questions. Consequently, a new regularization loss needs to be specifically designed for targeted unlearning.

## 4.2 OUR APPROACH

Intuitively, given a question in the retain set, the regularization loss for targeted unlearning should satisfy two objectives: 1) Reduce the probability of the rejection template. 2) Maintain the probability of the original answer. Thus, we propose the Answer Preservation (AP) loss as follows:

$$\mathcal{L}_{\text{AP}}(\mathcal{D}_{\text{R}}, \mathcal{D}_{\text{IDK}}; \theta) = -\frac{1}{\beta} \mathbb{E}_{(x,y)\sim\mathcal{D}_{\text{R}}, y'\sim\mathcal{D}_{\text{IDK}}} \left[ \log \sigma \left( -\beta \log \frac{p(y'|x;\theta)}{p(y|x;\theta)} \right) \right], \quad (8)$$

where $\sigma(\cdot)$ is the sigmoid function, $\beta$ is a hyper-parameter. Eq. (8) is somewhat similar in form to preference optimization (Zhang et al., 2024a; Rafailov et al., 2024), but it does not need a reference model and used for regularization rather than forgetting. We also perform gradient analysis on AP loss in Appendix C, and obtain the result as follows:

$$\nabla_\theta \mathcal{L}_{\text{AP}}(\theta) = \mathbb{E}_{\mathcal{D}_{\text{R}}, \mathcal{D}_{\text{IDK}}} \left[ W_\theta(x, y, y') \nabla_\theta \left( \log p(y'|x;\theta) - \log p(y|x;\theta) \right) \right]. \quad (9)$$

The $W_\theta(x, y, y') = 1/(1 + (\frac{p(y|x;\theta)}{p(y'|x;\theta)})^\beta)$ can be regarded as an adaptive gradient weight. Given a question $x$ in $\mathcal{D}_{\text{R}}$, in the early stage of unlearning process, where $p(y|x;\theta) \gg p(y'|x;\theta)$, we have $W_\theta(x, y, y') \ll 1$. As the unlearning proceeds, either a decrease in $p(y|x;\theta)$ or an increase in $p(y'|x;\theta)$ will result in a larger $W_\theta(x, y, y')$, thereby providing stronger regularization. The gradient of AP loss consists of two terms in addition to the adaptive weight. The first term is equivalent to GA on the rejection template, which satisfies the first objective. The second term is equivalent to GD on the original answer, which satisfies the second objective.

Finally, we combine AP as a regularization loss with IDK to get the approach IDK+AP:

$$\mathcal{L}_{\text{IDK+AP}}(\theta) = \mathcal{L}_{\text{IDK}}(\mathcal{D}_{\text{F}}; \theta) + \mathcal{L}_{\text{AP}}(\mathcal{D}_{\text{R}}, \mathcal{D}_{\text{IDK}}; \theta). \quad (10)$$

## 5 EXPERIMENTAL RESULTS

We show the main results for three different scenarios, namely fictitious unlearning, continual unlearning and real-world unlearning. More experimental results can be found in Appendix E.

Figure 6: Model Utility in continual forget01, forget05 and forget10 unlearning scenarios.

## 5.1 FICTITIOUS UNLEARNING SCENARIO

**Setup.** The standard TOFU benchmark (Maini et al., 2024) simulates an ideal scenario where the training data is fully accessible. It constructs a dataset with 200 fictitious authors, each containing 20 question-answer pairs. It has three levels of tasks, namely forget01, forget05 and forget10, to forget 1%, 5%, and 10% of the constructed data. The complement of each forget set serves as the retain set (i.e., neighbor set). It also provides two extra sets, namely Real Authors and World Facts, to evaluate the utility on general knowledge. Following Maini et al. (2024), we also calculate all metrics on these two extra sets and take the harmonic mean together with the metrics on the retain set as the final MU. We use the Llama2-chat-7B released by TOFU as the target model, which has been fine-tuned on the constructed data to ensure it can exactly gives answers to questions in TOFU.

**Results of untargeted unlearning.** Figure 4 shows the results of different untargeted unlearning methods. *Overall, our ME+GD achieves an surprising balance between MU and FE during unlearning, maintaining a stable MU with the highest FE.* For forget01, the MU of all methods is within an acceptable range, while baselines except ME+GD exhibit insufficient unlearning, i.e., the FE is relatively low. For forget05 and forget10, the GA-based method will quickly reduce MU in the early stage, but GA+GD can recover a certain MU when the FE reaches the bottleneck, which is also observed in (Maini et al., 2024). Although the NPO-based method can alleviate the reduce of MU, the FE is difficult to further improve, even if we increase the unlearning steps, as shown in Figure 9. In contrast, our ME+GD can continuously increase FE while maintaining the MU on all tasks.

**Results of targeted unlearning.** Figure 5 shows the results of different targeted unlearning methods. *Overall, only our IDK+AP maintains a stable MU on all three tasks.* For forget01, there is not much difference in MU between different methods, except that the DPO-based method has lower FE. However, as the size of forget set increases, the MU of baselines will quickly decrease and make the unlearned model overly ignorant, while our IDK+AP will still maintain a high MU.

**Additional results.** We provide more results, including: detailed quantitative results on TOFU (Appendix E.1), results of two variants, namely ME+KL and DPO+AP (Appendix E.2), an ablation on $\alpha$ in Eq. (7) (Appendix E.4), an ablation of whether to calculate ME loss over the questions (Appendix E.5), unlearning fine-tuning using LoRA (Hu et al., 2022) (Appendix E.8), and samples generated by different methods (Appendix G).

## 5.2 CONTINUAL UNLEARNING SCENARIO

**Task definition.** In practice, LLM unlearning may necessitate multiple processing of different unlearning requests, i.e., continual unlearning. To the best of our knowledge, there is currently no benchmark for continual LLM unlearning. To accommodate this scenario, we extend the TOFU benchmark by continually unlearning 1% (forget01), 5% (forget05), and 10% (forget10) of the constructed data for $N$ times respectively. For continual forget01 and continual forget05, we set $N$ to 10, meaning that 10% and 50% of the data are unlearned in total, respectively. For continual forget10, we set $N$ to 9 because at least 10% of the data needs to be retained for regularization and evaluation purposes in TOFU benchmark. When unlearning a certain subtask, all remaining unforgotten data is used as the retain set. Consequently, as the number of unlearning tasks increases, the retain set available for regularization will gradually decrease. After each unlearning subtask, we calculate corresponding MU and FE of the unlearned model.

**Main results.** In Figure 6, we compare the ability of different unlearning methods to maintain MU in this continual scenario, as this is considered to be more challenging (Gao et al., 2024; Dou et al., 2024; Shi et al., 2024a). For continual forget01 scenario, all baselines except IDK+GD will reduce MU to nearly zero on a certain subtask. In contrast, our ME+GD and IDK+AP can maintain MU at a higher value throughout the continual unlearning period. For continual forget05 and forget10 scenarios, ME+GD exhibits excellent ability to maintain MU, while the IDK+AP presents a downward

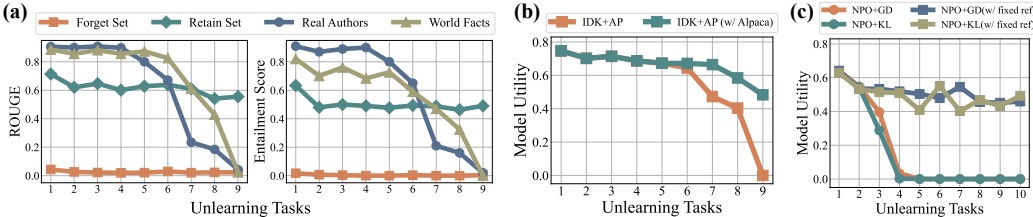

Figure 7: **(a)** ROUGE and Entailment Score of IDK+AP on different sets in the continual forget10 scenario. **(b)** Model Utility of IDK+AP with extra data in the continual forget10 scenario. **(c)** Model Utility of NPO-based methods with fixed reference in the continual forget01 scenario.

trend in the last few subtasks. When a total of 90% of data is unlearned, MU of IDK+AP also drops to nearly zero. In addition, the results of FE in continual unlearning scenario are shown in Figure 12.

**Effectiveness of AP Loss.** We investigate the phenomenon of MU decrease of IDK+AP in the last few subtasks in the continual forget10 scenario. As shown in Figure 7 (a), IDK+AP maintains stable ROUGE and ES on the retain set, *demonstrating the effectiveness of using AP for regularization in mitigating the unlearned model's ignorance on the retain set.* The decrease in MU is mainly due to the performance drop on two extra sets about general knowledge. We speculate that the gradual reduction of remaining samples in the retain set for regularization during continual unlearning may facilitate achieving $p(y|x;\theta) \gg p(y'|x;\theta)$, resulting in a smaller gradient weight in Eq. (9). We conducted an additional experiment in the continual forget10 scenario by randomly selecting samples from an unrelated dataset, Alpaca (Taori et al., 2023), to supplement the retain set, ensuring it always contains 2000 samples (i.e., starting supplementation after the 5th unlearning subtask). The results, shown in Figure 7 (b), indicate that this approach effectively mitigates the decrease in MU of the last few subtasks. Therefore, a critical issue for IDK+AP is how to reasonably expand the retain set when $|\mathcal{D}_R| \ll |\mathcal{D}_F|$ and we leave this to the future work.

**Failure analysis of NPO.** We observed a counterintuitive phenomenon: NPO-based methods can maintain acceptable MU for single forget05 and forget10 unlearning tasks, as shown in Section 5.1. However, in the continual forget01 scenario, the MU rapidly decreases to zero within a few subtasks, even though the total data to be unlearned is less. We attribute this phenomenon to the change of the reference model in continual scenarios, where the reference becomes the unlearned model from the previous subtask rather than the initial model. To further investigate, we conducted an additional experiment where we always fixed the reference model in NPO to the initial model. As shown in Figure 7 (c), this approach stabilizes MU throughout the continual unlearning tasks, although it remains lower than our two methods. Moreover, continuously maintaining the initial model still poses a privacy risk and violates the fundamental requirements of the Right to be Forgotten.

## 5.3 REAL-WORLD UNLEARNING SCENARIO

**Setup.** We consider a more realistic scenario where the knowledge to be unlearned is inherent in the target model and the training data are unknown. Liu et al. (2024c) identified several real-world individuals with deep memorization from Llama-3-8B-Instruct (Llama3) and provide 20 questions and golden answers for each individual. We first select 20 individuals as unlearning targets and use Llama3 to get responses for each question to construct the forget set. We then select different 40 individuals as the neighbor set, 20 of which were used for regularization during unlearning, consistent with the forget set size, and the other 20 are used to evaluate the MU. We also perform evaluation on five downstream tasks MMLU (Hendrycks et al., 2020), ARC-c (Clark et al., 2018), GSM8K (Cobbe et al., 2021), TriviaQA (Joshi et al., 2017), TruthfulQA(MC1) (Lin et al., 2021), which measure general ability, reasoning ability, arithmetic ability, factuality and truthfulness, respectively.

**Main results.** The results are shown in Table 2. *For untargeted unlearning,* our ME+GD method achieves the best performance on the unlearning task and is the only method capable of maintaining high MU and FE simultaneously. ME+GD exhibits superior overall performance on downstream tasks. In contrast, both GA-based and NPO-based methods struggle to maintain MU on the neighbor set during unlearning and significantly reduce average performance on downstream tasks. Notably, all methods except ME+GD significantly degrade performance on TriviaQA. We speculate that this is because the short question-answer format of TriviaQA closely resembles the format in the forget set, making it more susceptible to the unlearning process. *For targeted unlearning,* the main advantage of IDK+AP is its ability to preserve utility on the neighbor set. In contrast, the baseline methods treat the neighbor set and the forget set almost equally, leading to excessive ignorance on

Table 2: **Results of real-world unlearning scenario.** *Higher is better for all metrics.* Initial represent the original Llama3. Model Utility and Forget Efficacy are calculated on the neighbor set and forget set respectively.

| Method | Unlearning Task | | Downstream Tasks | | | | | |
|---|---|---|---|---|---|---|---|---|
| | Model Utility | Forget Efficacy | ARC-c | MMLU | TruthfulQA | TriviaQA | GSM8K | Avg. |
| Initial | 0.6145 | 0.3040 | 0.5657 | 0.6384 | 0.3611 | 0.5108 | 0.7551 | 0.5662 |
| **Untargeted Unlearning** | | | | | | | | |
| GA+GD | 0.2436 | 0.8969 | 0.5137 | 0.5880 | **0.3929** | 0.0744 | 0.2714 | 0.3681 |
| GA+KL | 0.1281 | 0.8556 | 0.4684 | 0.5839 | 0.2546 | 0.0079 | 0.2403 | 0.3110 |
| NPO+GD | 0.2144 | 0.6756 | 0.3840 | 0.5349 | 0.3415 | 0.0000 | 0.6929 | 0.3907 |
| NPO+KL | 0.1546 | 0.7000 | 0.3780 | 0.5180 | 0.3366 | 0.0000 | 0.6710 | 0.3807 |
| ME+GD(Ours) | **0.4901** | **0.9312** | **0.5299** | **0.6248** | 0.3121 | **0.4851** | **0.6952** | **0.5294** |
| **Targeted Unlearning** | | | | | | | | |
| DPO+GD | 0.0000 | 0.8574 | 0.5094 | 0.6216 | 0.3182 | 0.0856 | 0.7248 | 0.4519 |
| DPO+KL | 0.0000 | **0.8624** | 0.5068 | 0.6200 | 0.3146 | 0.0804 | 0.7218 | 0.4487 |
| IDK+GD | 0.0000 | 0.8210 | 0.5247 | 0.6248 | **0.3244** | 0.2477 | **0.7453** | 0.4934 |
| IDK+AP(Ours) | **0.5311** | 0.8244 | **0.5341** | 0.6204 | 0.2705 | **0.3360** | 0.7324 | **0.4987** |

the neighbor set. Our IDK+AP achieves the best average performance on downstream tasks. More detailed results are shown in Appendix E.7 and Appendix E.8.

# 6 RELATED WORK

**Memorization concerns of LLMs.** LLMs have demonstrated powerful capabilities through learning from extensive corpora. However, numerous studies have shown that they may inadvertently memorize information from training corpus (Huang et al., 2022; Carlini et al., 2023; Staab et al., 2024; Ippolito et al., 2023), which poses significant privacy and copyright concerns. With the enactment and widespread adoption of regulations such as the European Union's General Data Protection Regulation (GDPR) (Regulation, 2016) and the California Consumer Privacy Act (Pardau, 2018), which include provisions for the Right to be Forgotten (Dang, 2021), users may request the removal of specified content from LLM-driven applications. To protect users' legitimate interests and mitigate the risk of legal repercussions, these models must comply with such requests and avoid providing relevant responses (Si et al., 2023). Retraining the model on the filtered data is straightforward but impractical, particularly in the context of LLMs (Kandpal et al., 2022; Liu et al., 2024b).

**Machine unlearning for LLMs.** Initially developed for classification tasks (Bourtoule et al., 2021), machine unlearning has recently been applied to LLMs. Mainstream methods primarily rely on parameter optimization (Jang et al., 2023; Yao et al., 2023; Maini et al., 2024; Wang et al., 2024b; Li et al., 2024; Yao et al., 2024; Ishibashi & Shimodaira, 2023; Gu et al., 2024; Zhang et al., 2024a; Lu et al., 2024; Jia et al., 2024; Tian et al., 2024; Liu et al., 2024c; Choi et al., 2024; Tang et al., 2024; Tamirisa et al., 2024). This typically involves fine-tuning the model on a forget set to produce an unlearned version. Users may trust this paradigm more because it internally modifies the model's parameters and mechanisms. However, such methods are more likely to harm the overall performance. Researchers also introduce other techniques for LLM unlearning, including contrastive decoding (Eldan & Russinovich, 2023; Huang et al., 2024; Wang et al., 2024a; Ji et al., 2024; Dong et al., 2024), task vectors (Dou et al., 2024; Liu et al., 2024d), in-context learning (Pawelczyk et al., 2023; Muresanu et al., 2024; Thaker et al., 2024), and input processing and detection (Bhaila et al., 2024; Gao et al., 2024; Liu et al., 2024a). However, most of these methods require preserving the original model's parameters, which may still pose privacy concerns. The inability to truly remove user's specified information could also lead to potential litigation risks in the future.

# 7 CONCLUSION

In this paper, we discuss several issues in LLM unlearning and provide our insights on possible approaches. To address the issue of inadequate evaluation on the unlearned model's output, we introduce three additional metrics to evaluate token diversity, sentence semantics, and factual correctness. We then categorize previous methods into untargeted and targeted based on whether the response to the forget set is specified. For untargeted unlearning, we discuss that the behavior it attempts to approximate is unpredictable and may involve hallucinations, and adopt the objective maximizing entropy. For targeted unlearning, we analyze that existing regularization is insufficient and incorporating the answer preservation loss as a regularization term. Extensive experiments across various scenarios demonstrate the effectiveness of our approaches.

ETHICAL STATEMENT

In light of emerging regulations, the removal of user-specified data from large language models (LLMs) is increasingly recognized as a critical component of responsible and ethical AI development. This study further investigates the application of machine unlearning techniques to LLMs. The datasets utilized for evaluation are publicly available and implemented within the intended use. We hope this study to advance research and literature on machine unlearning for LLMs.

ACKNOWLEDGMENTS

This work was supported in part by the National Natural Science Foundation of China under Grant 62121002, 62472398 and U2336206.

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

# A PROMPT FOR HALLUCINATION JUDGMENT

We use the prompt in Figure 8 for GPT-4o to determine whether the output of the model is considered a hallucination.

---

**Prompt for hallucination judgment:**

You will be given a question, a reference answer and a generated answer. Your task is to provide a judgment on whether the generated answer is considered a "hallucination".

Here are the criteria for "hallucination":
- The generated answer MUST give a direct response to the question.
- The generated answer MUST be inconsistent with the facts in the reference answer.
NOTE: If the generated answer apologizes for lack of relevant knowledge or admits not knowing the correct answer, it is NOT considered a "hallucination".

Here are the question, reference answer, and generated answer:
Question: {QUESTION}
Reference Answer: {GROUND TRUTH}
Generated Answer: {OUTPUT}

Provide your judgment, if the generated answer is considered a "hallucination", return "YES". Otherwise, simply return "NO".

---

Figure 8: The full prompt for GPT-4o to give a judgment on hallucination.

# B DERIVATION OF MAXIMIZING ENTROPY LOSS

In Section 3, we propose to minimize the KL divergence between the predicted distribution for each token and a uniform distribution with vocabulary size, as follows:

$$\mathcal{L}(\mathcal{D}_{\mathrm{F}};\theta) = \mathbb{E}_{(x,y)\sim\mathcal{D}_F}\left[\frac{1}{T}\sum_{t=1}^{T}\mathrm{KL}(P_t\|\mathcal{U}_{[K]})\right], \tag{11}$$

where $P_t = p(x'_t|x'_{<t};\theta)$ is the predicted probability of the model for the $t$-th token in $x' = x \circ y$ and $\mathcal{U}_{[K]}$ denotes a uniform distribution over a vocabulary of size $K$, where each value is $1/K$. Consider a certain sample, we can further derive it as follows:

$$\begin{aligned}
\frac{1}{T}\sum_{t=1}^{T}KL(P_t \| \mathcal{U}_{[K]}) &= \frac{1}{T}\sum_{t=1}^{T}\sum_{j=1}^{K}P_t(j)\log\left(\frac{P_t(j)}{\mathcal{U}_{[K]}(j)}\right) \\
&= \frac{1}{T}\sum_{t=1}^{T}\sum_{j=1}^{K}P_t(j)\log(P_t(j)\cdot K) \\
&= \frac{1}{T}\sum_{t=1}^{T}\left[\sum_{j=1}^{K}P_t(j)\log P_t(j) + \sum_{j=1}^{K}P_t(j)\log K\right] \\
&= \frac{1}{T}\sum_{t=1}^{T}\left[\sum_{j=1}^{K}P_t(j)\log P_t(j) + \log K\right] \\
&= \frac{1}{T}\sum_{t=1}^{T}\left[-H(P_t) + \log K\right] \\
&= -\frac{1}{T}\sum_{t=1}^{T}H(P_t) + \log K
\end{aligned} \tag{12}$$

where $P_t(j)$ is the $j$-th element of $P_t$ and $H(\cdot)$ is the entropy of a given distribution. Therefore, minimizing Eq. (11) is actually equivalent to maximizing the entropy of predicted distribution for each next token.

## C  GRADIENT ANALYSIS OF AP LOSS

In Section 4, we use the AP loss as regularization for targeted unlearning as follows:

$$\mathcal{L}_{\mathrm{AP}}(\mathcal{D}_{\mathrm{R}}, \mathcal{D}_{\mathrm{IDK}}; \theta) = -\frac{1}{\beta}\mathbb{E}_{(x,y)\sim\mathcal{D}_{\mathrm{R}}, y'\sim\mathcal{D}_{\mathrm{IDK}}}\left[\log\sigma\left(-\beta\log\frac{p(y'|x;\theta)}{p(y|x;\theta)}\right)\right], \tag{13}$$

where $\sigma(\cdot)$ is the sigmoid function, $\beta$ is a hyper-parameter. Let $M_\theta = \log\frac{p(y'|x;\theta)}{p(y|x;\theta)}$ where $(x,y) \sim \mathcal{D}_{\mathrm{R}}$ and $y' \sim \mathcal{D}_{\mathrm{IDK}}$, we can perform the gradient analysis on AP loss as follows:

$$
\begin{aligned}
\nabla_\theta\mathcal{L}_{\mathrm{AP}}(\theta) &= -\frac{1}{\beta}\mathbb{E}_{\mathcal{D}_{\mathrm{R}},\mathcal{D}_{\mathrm{IDK}}}\left[\nabla_\theta\log\sigma\left(-\beta M_\theta\right)\right]\\
&= \frac{1}{\beta}\mathbb{E}_{\mathcal{D}_{\mathrm{R}},\mathcal{D}_{\mathrm{IDK}}}\left[\nabla_\theta\log\left(1+\exp(\beta M_\theta)\right)\right]\\
&= \frac{1}{\beta}\mathbb{E}_{\mathcal{D}_{\mathrm{R}},\mathcal{D}_{\mathrm{IDK}}}\left[\frac{\beta\exp(\beta M_\theta)}{1+\exp(\beta M_\theta)}\nabla_\theta M_\theta\right]\\
&= \mathbb{E}_{\mathcal{D}_{\mathrm{R}},\mathcal{D}_{\mathrm{IDK}}}\left[W_\theta(x,y,y')\cdot\nabla_\theta\log\frac{p(y'|x;\theta)}{p(y|x;\theta)}\right]\\
&= \mathbb{E}_{\mathcal{D}_{\mathrm{R}},\mathcal{D}_{\mathrm{IDK}}}\left[W_\theta(x,y,y')\nabla_\theta\log p(y'|x;\theta)\right]\\
&\quad + \mathbb{E}_{\mathcal{D}_{\mathrm{R}},\mathcal{D}_{\mathrm{IDK}}}\left[W_\theta(x,y,y')\nabla_\theta\left[-\log p(y|x;\theta)\right]\right]\\
&= \mathbb{E}_{\mathcal{D}_{\mathrm{R}},\mathcal{D}_{\mathrm{IDK}}}\left[W_\theta(x,y,y')\nabla_\theta\left(\log p(y'|x;\theta) - \log p(y|x;\theta)\right)\right]
\end{aligned}
\tag{14}
$$

The $W_\theta(x,y,y') = 1/(1 + (\frac{p(y|x;\theta)}{p(y'|x;\theta)})^\beta)$ can be regarded as an adaptive gradient weight. Given a question $x$ in $\mathcal{D}_{\mathrm{R}}$, in the early stage of the unlearning, where $p(y|x;\theta) \gg p(y'|x;\theta)$, we have $W_\theta(x,y,y') \ll 1$. As the unlearning process proceeds, either a decrease in $p(y|x;\theta)$ or an increase in $p(y'|x;\theta)$ will result in a larger $W_\theta(x,y,y')$, thereby providing stronger regularization. It can be seen that the gradient of AP consists of two terms, the first term is equivalent to GA on the rejection template, and the second term is equivalent to GD on the original answer, which meets our requirements for the regularization in targeted unlearning .

## D  IMPLEMENTATION DETAILS

**TOFU dataset.** For the experiments on the TOFU dataset, we use the fine-tuned Llama2-chat-7B model released by the original paper (Maini et al., 2024) as the target model.[1] All experiments are conducted on two NVIDIA A100 GPUs with 40GB of memory. We follow the TOFU repository[2] and utilize DeepSpeed with ZeRO3 to reduce memory costs. Following the configuration in (Maini et al., 2024), we employ the AdamW optimizer with a weight decay of $0.01$, a learning rate of $1 \times 10^{-5}$, and an effective batch size of 32 for all experiments, consistent with the settings in (Maini et al., 2024; Zhang et al., 2024a). During unlearning, we fine-tune for 5 epochs, using a linear warm-up learning rate in the first epoch and a linearly decaying learning rate in the subsequent epochs. Following the setup in (Maini et al., 2024; Zhang et al., 2024a), we randomly sample up to 300 question-answer pairs from the dataset for evaluation to improve efficiency. The $\beta$ in NPO and AP is set to $0.1$. The parameter $\alpha$ in MG+GD is set to $0.1$ in Section 5.1 and is set to $1.0$ in Section 5.2, because the difficulty of different unlearning tasks in continual scenes may different, we give greater unlearning strength.

**Real-world individuals.** Follow (Liu et al., 2024c), we use Llama-3-8B-Instruct[3] as the target model. We use the repository[4] of lm-evaluation-harness (Gao et al., 2023) to evaluate downstream

---

[1] https://huggingface.co/locuslab/tofu_ft_llama2-7b

[2] https://github.com/locuslab/tofu

[3] https://huggingface.co/meta-llama/Meta-Llama-3-8B-Instruct

[4] https://github.com/EleutherAI/lm-evaluation-harness

tasks with default configurations. For ME+GD, we set the number of unlearning epochs to 5, the learning rate to $5 \times 10^{-6}$, and $\alpha$ to 0.5. For IDK+AP, we set the number of unlearning epochs to 5 and the learning rate to $1 \times 10^{-5}$. For the baseline methods, we tune the number of unlearning epochs from $\{3, 5\}$ and the learning rate from $\{2 \times 10^{-6}, 5 \times 10^{-6}, 1 \times 10^{-5}\}$. Given the need for unlearning to be generalizable, we compute unlearning metrics using golden answers rather than original answers in the forget set. We report the best results considering both the unlearning task and downstream tasks. Other configurations are consistent with TOFU dataset.

**Evaluation metrics.** The six general metrics, namely *R*, *P*, *TR*, *TE*, *CS* and *ES*, only reflect the performance of the model on a certain set, with higher indicating better performance. For unlearning tasks, the unlearned model should have good performance on retain set and general knowledge, but poor performance on the forget set. We require the two aggregated metrics, i.e., *MU* and *FE*, can range from 0 and 1, with higher indicating better results in the unlearning task We calculate the *R*, *P* and *TR* according to previous work (Maini et al., 2024; Zhang et al., 2024a; Ji et al., 2024). In the TOFU benchmark, the calculation of *P* is different on the Real Author set and the World Fact set, because these two sets only provide a single word as the ground truth answer. Specifically, each question $q$ is treated as a multiple choice question associated with choices $\{y_1, \ldots, y_n\}$. Assume that $y_1$ is the only correct answer, then the *P* is computed as $p(y_1|q) / \sum_{i=1}^{n} p(y_i|q)$.

For the *TR*, we report $\max(0, 1 - \text{TR})$ on the retain set and $1 - \min(\text{TR}, 1/\text{TR})$ on the forget set. We only use *TE* to measure the utility of the unlearned model on the retain set and general knowledge because the unlearned model may output random content as a result of untargeted unlearning. We obtain the checkpoints of the Sentence-BERT[5] and the NLI model[6] used to calculate *CS* and *ES* respectively from the HuggingFace. When calculating *ES* on TOFU dataset, we consider if $y_R \Rightarrow g(x_R; \theta_u)$ on the retain set, which evaluate whether the output contains only correct information that can be inferred from ground truth and no made-up content. While on the forget set, we consider if $g(x_F; \theta_u) \Rightarrow y_F$, which evaluate whether any information in the ground truth can be correctly inferred from the output, indicating potential information leakage. When calculating *ES* on real-world individuals, we consider $g(x; \theta_u) \Rightarrow y$ on both the forget set and the neighbor set, cause the golden answer for evaluation is usually too short for relationship inference, as shown in Table 3.

Table 3: Samples of forget set in real-world unlearning scenario.

| Question | What nationality is Dario Argento? |
|---|---|
| Answer | Dario Argento is an Italian film director, screenwriter, and producer. He was born on September 7, 1940, in Rome, Italy. |
| Golden Answer | Dario Argento is Italian. |

# E   MORE EXPERIMENTAL RESULTS

## E.1   DETAILED RESULTS ON THE TOFU BENCHMARK

In Table 4, we present the final quantitative results of various unlearning methods on the TOFU benchmark, corresponding to Figure 4 and Figure 5 in Section 5.1. Additionally, detailed results for each metric are provided in Table 5 and Table 6.

## E.2   RESULTS OF ME+KL AND DPO+AP ON THE TOFU BENCHMARK

We evaluate two variations of our methods on the TOFU benchmark: ME+KL and DPO+AP. The results are shown in Table 7. ME+KL can slightly increase FE, but MU is significantly decreased. DPO+AP can maintain a high MU, demonstrating the effectiveness of AP for targeted unlearning.

## E.3   INCREASE THE NUMBER OF UNLEARNING STEPS ON TOFU

We perform unlearning for 10 epochs on the three tasks in TOFU, and compare the changing trends of MU and FE during the entire process using different untargeted methods. As shown in Figure 9,

---

[5]https://huggingface.co/sentence-transformers/paraphrase-MiniLM-L6-v2

[6]https://huggingface.co/sileod/deberta-v3-base-tasksource-nli. We observe that this model seems to have some issues with processing sentence consist of meaningless characters. Therefore, we direct label the pair as "not entailment", if their ROUGE is less than 0.1.

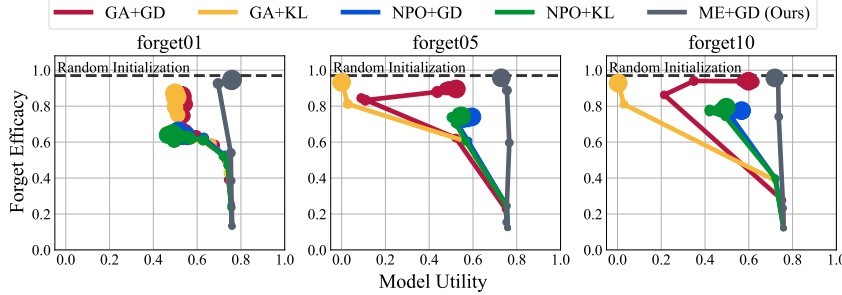

Figure 9: Forget Efficacy versus Model Utility of untargeted unlearning on different tasks of TOFU *when unlearning for 10 epochs.* The relative size of the markers indicates the epoch of unlearning.

Table 4: Results of different unlearning methods on the TOFU benchmark. MU and FE represent Model Utility and Forget Efficacy respectively and we indicate the best and second best results in bold and underline respectively.

| Method | forget01 | | | forget05 | | | forget10 | | |
|---|---|---|---|---|---|---|---|---|---|
| | MU | FE | Avg. | MU | FE | Avg. | MU | FE | Avg. |
| GA+GD | 0.6696 | 0.5908 | 0.6302 | 0.0000 | 0.8772 | 0.4386 | 0.5592 | 0.9346 | 0.7469 |
| GA+KL | 0.6478 | 0.6030 | 0.6254 | 0.0000 | 0.9303 | 0.4651 | 0.0000 | 0.9020 | 0.4510 |
| NPO+GD | 0.6414 | 0.6109 | 0.6262 | 0.5465 | 0.6921 | 0.6193 | 0.5648 | 0.7668 | 0.6658 |
| NPO+KL | 0.6289 | 0.6065 | 0.6177 | 0.5231 | 0.7089 | 0.6160 | 0.4412 | 0.7680 | 0.6046 |
| ME+GD(Ours) | 0.7271 | 0.9204 | 0.8237 | 0.7472 | 0.9313 | 0.8392 | 0.7357 | 0.9489 | 0.8423 |
| DPO+GD | 0.7564 | 0.5335 | 0.6450 | 0.0000 | 0.8243 | 0.4122 | 0.0000 | 0.8041 | 0.4021 |
| DPO+KL | 0.7577 | 0.5383 | 0.6480 | 0.0000 | 0.8339 | 0.4169 | 0.0000 | 0.8421 | 0.4211 |
| IDK+GD | 0.6705 | 0.7697 | 0.7201 | 0.0000 | 0.7952 | 0.3976 | 0.0576 | 0.7603 | 0.4090 |
| IDK+AP(Ours) | 0.7580 | 0.7625 | 0.7603 | 0.7529 | 0.7479 | 0.7504 | 0.7471 | 0.7433 | 0.7452 |

the FE of NPO-based method will not further improve with the increase of unlearning step. This is because its gradient will decrease as the conditional probability on the forget set decreases (Zhang et al., 2024a), making the unlearning process to stop prematurely. However, the unlearning behavior of a model should consider multiple perspectives as mentioned in Section 2.2. In contrast, our ME+GD can continuously increase FE while maintaining a stable MU on all tasks. For ME+GD, increasing the number of unlearning steps after FE reaches the bottleneck will not damage MU. In addition, we find that GA+GD can also recover a certain MU by performing more unlearning steps on forget05, but this recovery is capped and still significantly lower than our method.

### E.4 DIFFERENT UNLEARNING STRENGTH IN ME+GD

In Eq. (7), the hyper-parameter $\alpha$ in ME+GD balances the weights of the forget loss and regularization loss, allowing for different levels of unlearning strength. As shown in Figure 10, a too small value of $\alpha$ leads to diminished forget efficacy. As $\alpha$ increases, forget efficacy gradually improves and stabilizes after reaching 0.1. Model utility is generally insensitive to changes in $\alpha$; however, in the forget01 task, an excessively large $\alpha$ results in a slight decrease in model utility.

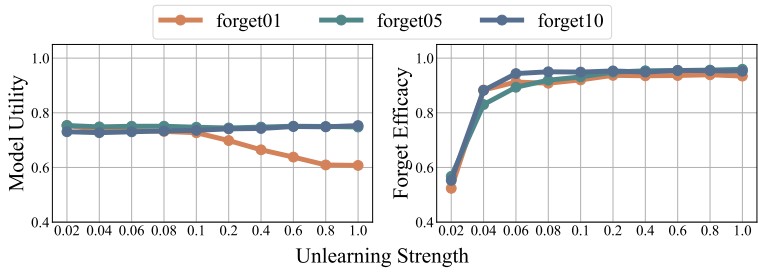

Figure 10: Results of ME+GD with different unlearning strength on TOFU benchmark.

Table 5: Detailed results for each metric on the retain set and the forget set for three tasks in the TOFU benchmark.

| Task | Method | Retain Set | | | | | | Forget Set | | | | |
|---|---|---|---|---|---|---|---|---|---|---|---|---|
| | | R ↑ | P ↑ | TR ↑ | TE ↑ | CS ↑ | ES ↑ | R ↓ | P ↓ | TR ↓ | CS ↓ | ES ↓ |
| forget01 | GA+GD | 0.8137 | 0.8785 | 0.4959 | 0.9539 | 0.9147 | 0.4500 | 0.4235 | 0.0913 | 0.4625 | 0.7687 | 0.3000 |
| | GA+KL | 0.8508 | 0.8415 | 0.4960 | 0.9489 | 0.9149 | 0.3667 | 0.4519 | 0.0870 | 0.4394 | 0.7816 | 0.2250 |
| | NPO+GD | 0.8669 | 0.8428 | 0.4964 | 0.9475 | 0.9212 | 0.3400 | 0.4427 | 0.0989 | 0.3624 | 0.7664 | 0.2750 |
| | NPO+KL | 0.8634 | 0.8133 | 0.4961 | 0.9457 | 0.9154 | 0.3100 | 0.4529 | 0.0976 | 0.3664 | 0.7756 | 0.2750 |
| | ME+GD(Ours) | 0.7805 | 0.8896 | 0.4500 | 0.9686 | 0.9058 | 0.6333 | 0.0260 | 0.0042 | 0.2551 | 0.1126 | 0.0000 |
| | DPO+GD | 0.8888 | 0.9658 | 0.4566 | 0.9731 | 0.9577 | 0.9467 | 0.3608 | 0.8397 | 0.4060 | 0.6008 | 0.1250 |
| | DPO+KL | 0.8858 | 0.9656 | 0.4570 | 0.9734 | 0.9569 | 0.9500 | 0.3627 | 0.8397 | 0.4066 | 0.5993 | 0.1000 |
| | IDK+GD | 0.4726 | 0.9370 | 0.4556 | 0.9874 | 0.5541 | 0.5200 | 0.0095 | 0.7157 | 0.3993 | 0.0268 | 0.0000 |
| | IDK+AP(Ours) | 0.8758 | 0.9701 | 0.4595 | 0.9737 | 0.9519 | 0.9233 | 0.0128 | 0.7239 | 0.4011 | 0.0498 | 0.0000 |
| forget05 | GA+GD | 0.0293 | 0.0004 | 0.5805 | 0.0087 | 0.0300 | 0.0733 | 0.0101 | 0.0000 | 0.5680 | 0.0307 | 0.0050 |
| | GA+KL | 0.0246 | 0.0000 | 0.2578 | 0.0837 | 0.1015 | 0.0333 | 0.0047 | 0.0000 | 0.2570 | 0.0870 | 0.0000 |
| | NPO+GD | 0.5321 | 0.3168 | 0.4508 | 0.8466 | 0.7359 | 0.2867 | 0.3781 | 0.0899 | 0.3167 | 0.5749 | 0.1800 |
| | NPO+KL | 0.5221 | 0.1874 | 0.4447 | 0.8026 | 0.7014 | 0.4400 | 0.3710 | 0.0596 | 0.3186 | 0.5515 | 0.1550 |
| | ME+GD(Ours) | 0.8813 | 0.9424 | 0.4477 | 0.9682 | 0.9447 | 0.8167 | 0.0464 | 0.0161 | 0.1759 | 0.1052 | 0.0000 |
| | DPO+GD | 0.0055 | 0.6002 | 0.3761 | 0.9999 | 0.0556 | 0.0000 | 0.0011 | 0.4856 | 0.3437 | 0.0481 | 0.0000 |
| | DPO+KL | 0.0023 | 0.5501 | 0.3682 | 1.0000 | 0.0524 | 0.0000 | 0.0011 | 0.4463 | 0.3350 | 0.0481 | 0.0000 |
| | IDK+GD | 0.0126 | 0.7404 | 0.4041 | 0.9487 | 0.0541 | 0.0033 | 0.0138 | 0.5961 | 0.3704 | 0.0436 | 0.0000 |
| | IDK+AP(Ours) | 0.7569 | 0.9079 | 0.4425 | 0.9673 | 0.8994 | 0.6467 | 0.0307 | 0.7084 | 0.4226 | 0.0888 | 0.0100 |
| forget10 | GA+GD | 0.3754 | 0.4350 | 0.5043 | 0.8623 | 0.6485 | 0.3567 | 0.0098 | 0.0000 | 0.2629 | 0.0445 | 0.0100 |
| | GA+KL | 0.0000 | 0.0000 | 0.1636 | 0.0000 | 0.0620 | 0.0000 | 0.0000 | 0.0000 | 0.4223 | 0.0677 | 0.0000 |
| | NPO+GD | 0.4245 | 0.3299 | 0.3528 | 0.7022 | 0.6100 | 0.6133 | 0.2429 | 0.1254 | 0.2822 | 0.4187 | 0.0967 |
| | NPO+KL | 0.3211 | 0.1142 | 0.3148 | 0.5331 | 0.4699 | 0.7233 | 0.2477 | 0.0664 | 0.2803 | 0.4157 | 0.1500 |
| | ME+GD(Ours) | 0.8574 | 0.9479 | 0.4511 | 0.9684 | 0.9412 | 0.7833 | 0.0392 | 0.0098 | 0.1122 | 0.0908 | 0.0033 |
| | DPO+GD | 0.0088 | 0.6149 | 0.3750 | 0.9999 | 0.0930 | 0.0000 | 0.0050 | 0.5442 | 0.3470 | 0.0833 | 0.0000 |
| | DPO+KL | 0.0031 | 0.4779 | 0.3482 | 1.0000 | 0.0572 | 0.0000 | 0.0013 | 0.4196 | 0.3206 | 0.0479 | 0.0000 |
| | IDK+GD | 0.1417 | 0.8341 | 0.4266 | 0.9743 | 0.2279 | 0.1367 | 0.0109 | 0.7359 | 0.4065 | 0.0451 | 0.0000 |
| | IDK+AP(Ours) | 0.7141 | 0.8922 | 0.4618 | 0.9685 | 0.8878 | 0.6333 | 0.0433 | 0.6960 | 0.4451 | 0.0823 | 0.0167 |

Table 6: Detailed results for each metric on the real authors set and the word facts set for three tasks in the TOFU benchmark.

| Task | Method | Real Authors Set | | | | | | World Facts Set | | | | | |
|---|---|---|---|---|---|---|---|---|---|---|---|---|---|
| | | R ↑ | P ↑ | TR ↑ | TE ↑ | CS ↑ | ES ↑ | R ↑ | P ↑ | TR ↑ | TE ↑ | CS ↑ | ES ↑ |
| forget01 | GA+GD | 0.9030 | 0.4029 | 0.5410 | 0.9739 | 0.9346 | 0.8600 | 0.8689 | 0.3910 | 0.5302 | 0.9404 | 0.9250 | 0.5897 |
| | GA+KL | 0.8950 | 0.4025 | 0.5358 | 0.9647 | 0.9129 | 0.7600 | 0.8818 | 0.3936 | 0.5281 | 0.9348 | 0.9155 | 0.5299 |
| | NPO+GD | 0.9150 | 0.3978 | 0.5237 | 0.9559 | 0.9013 | 0.7800 | 0.8775 | 0.3906 | 0.5231 | 0.9294 | 0.9097 | 0.5299 |
| | NPO+KL | 0.9150 | 0.3960 | 0.5245 | 0.9506 | 0.8909 | 0.7600 | 0.8775 | 0.3909 | 0.5206 | 0.9266 | 0.8983 | 0.4957 |
| | ME+GD(Ours) | 0.8697 | 0.5079 | 0.6544 | 0.9835 | 0.9379 | 0.8300 | 0.8575 | 0.4649 | 0.6142 | 0.9540 | 0.9444 | 0.7009 |
| | DPO+GD | 0.9263 | 0.4893 | 0.6329 | 0.9867 | 0.9601 | 0.9200 | 0.8718 | 0.4567 | 0.5721 | 0.9667 | 0.9498 | 0.7692 |
| | DPO+KL | 0.9263 | 0.4891 | 0.6335 | 0.9868 | 0.9594 | 0.9200 | 0.8803 | 0.4565 | 0.5709 | 0.9662 | 0.9534 | 0.7863 |
| | IDK+GD | 0.8663 | 0.4739 | 0.6117 | 0.9887 | 0.9039 | 0.8500 | 0.8661 | 0.4458 | 0.5635 | 0.9676 | 0.9483 | 0.7863 |
| | IDK+AP(Ours) | 0.9263 | 0.4923 | 0.6355 | 0.9873 | 0.9629 | 0.9100 | 0.8746 | 0.4544 | 0.5762 | 0.9659 | 0.9601 | 0.8034 |
| forget05 | GA+GD | 0.0000 | 0.4263 | 0.6261 | 0.0311 | 0.0828 | 0.0000 | 0.4141 | 0.4567 | 0.5933 | 0.1030 | 0.2055 | 0.4188 |
| | GA+KL | 0.0503 | 0.4926 | 0.6478 | 0.1492 | 0.1338 | 0.0700 | 0.1624 | 0.4222 | 0.5782 | 0.1621 | 0.2020 | 0.1453 |
| | NPO+GD | 0.9243 | 0.3822 | 0.4851 | 0.8639 | 0.7893 | 0.7900 | 0.9117 | 0.4108 | 0.5331 | 0.8717 | 0.8229 | 0.4103 |
| | NPO+KL | 0.9077 | 0.3640 | 0.4602 | 0.8393 | 0.7787 | 0.7500 | 0.9010 | 0.3966 | 0.5178 | 0.8526 | 0.8090 | 0.4615 |
| | ME+GD(Ours) | 0.9050 | 0.4878 | 0.6358 | 0.9859 | 0.9563 | 0.8700 | 0.8832 | 0.4569 | 0.5911 | 0.9596 | 0.9628 | 0.7607 |
| | DPO+GD | 0.0053 | 0.4418 | 0.5802 | 1.0000 | 0.0274 | 0.0000 | 0.2650 | 0.4407 | 0.5525 | 0.9886 | 0.2874 | 0.2650 |
| | DPO+KL | 0.0053 | 0.4366 | 0.5727 | 1.0000 | 0.0274 | 0.0000 | 0.1752 | 0.4388 | 0.5477 | 0.9928 | 0.1930 | 0.1709 |
| | IDK+GD | 0.0053 | 0.4494 | 0.5843 | 0.9595 | 0.0255 | 0.0000 | 0.0000 | 0.4353 | 0.5414 | 0.9738 | 0.0107 | 0.0000 |
| | IDK+AP(Ours) | 0.8973 | 0.5704 | 0.7361 | 0.9855 | 0.9324 | 0.9000 | 0.8860 | 0.5042 | 0.6239 | 0.9619 | 0.9359 | 0.7778 |
| forget10 | GA+GD | 0.5678 | 0.6416 | 0.7983 | 0.7257 | 0.5428 | 0.3900 | 0.8205 | 0.5432 | 0.6843 | 0.8697 | 0.7868 | 0.4103 |
| | GA+KL | 0.0000 | 0.2446 | 0.4465 | 0.0000 | 0.0373 | 0.0000 | 0.0000 | 0.2590 | 0.4233 | 0.0000 | 0.0164 | 0.0000 |
| | NPO+GD | 0.9350 | 0.4456 | 0.5889 | 0.8161 | 0.7085 | 0.6300 | 0.8889 | 0.4360 | 0.5721 | 0.8055 | 0.7738 | 0.4957 |
| | NPO+KL | 0.7130 | 0.4381 | 0.5684 | 0.5588 | 0.5684 | 0.6900 | 0.7956 | 0.4146 | 0.5624 | 0.6111 | 0.6705 | 0.7436 |
| | ME+GD(Ours) | 0.8933 | 0.4709 | 0.6087 | 0.9846 | 0.9610 | 0.8600 | 0.9060 | 0.4397 | 0.5652 | 0.9619 | 0.9535 | 0.7607 |
| | DPO+GD | 0.0053 | 0.4234 | 0.5491 | 1.0000 | 0.0275 | 0.0000 | 0.1724 | 0.4198 | 0.5176 | 0.9932 | 0.1965 | 0.1709 |
| | DPO+KL | 0.0053 | 0.4118 | 0.5309 | 1.0000 | 0.0274 | 0.0000 | 0.0000 | 0.4115 | 0.5053 | 1.0000 | 0.0115 | 0.0000 |
| | IDK+GD | 0.0153 | 0.4517 | 0.5832 | 0.9995 | 0.0372 | 0.0100 | 0.0285 | 0.4245 | 0.5347 | 0.9951 | 0.0446 | 0.0342 |
| | IDK+AP(Ours) | 0.9063 | 0.5748 | 0.7209 | 0.9847 | 0.9488 | 0.9100 | 0.8860 | 0.4735 | 0.5794 | 0.9631 | 0.9569 | 0.8205 |

## E.5 ABLATION STUDY OF CALCULATING THE ME LOSS ON QUESTIONS

Most previous work follow the typical instruction tuning paradigm when calculating the forget loss in unlearning fine-tuning (Maini et al., 2024; Zhang et al., 2024a; Jia et al., 2024; Yao et al., 2023), that is, *masking the instruction (question) part of each sample when calculating the forget loss.* We

Table 7: Results of variants of our methods on the TOFU benchmark.

| Method | forget01 | | | forget05 | | | forget10 | | |
|---|---|---|---|---|---|---|---|---|---|
| | MU | FE | Avg. | MU | FE | Avg. | MU | FE | Avg. |
| ME+KL | 0.5074 | **0.9585** | 0.7329 | 0.6213 | **0.9715** | 0.7964 | 0.6418 | **0.9656** | 0.8037 |
| ME+GD | **0.7271** | 0.9204 | **0.8237** | **0.7472** | 0.9313 | **0.8392** | **0.7357** | 0.9489 | **0.8423** |
| DPO+AP | 0.7591 | 0.4734 | 0.6162 | 0.7173 | 0.5455 | 0.6314 | 0.6990 | 0.5380 | 0.6185 |
| IDK+AP | **0.7580** | **0.7625** | **0.7603** | **0.7529** | **0.7479** | **0.7504** | **0.7471** | **0.7433** | **0.7452** |

conducted an ablation experiment on ME+GD to study the influence of questions are masked when calculating the forget loss. The results are presented in Figure 11. It can be seen that when using this question masking operation, the performance of the unlearned model on the retain set becomes more unstable.

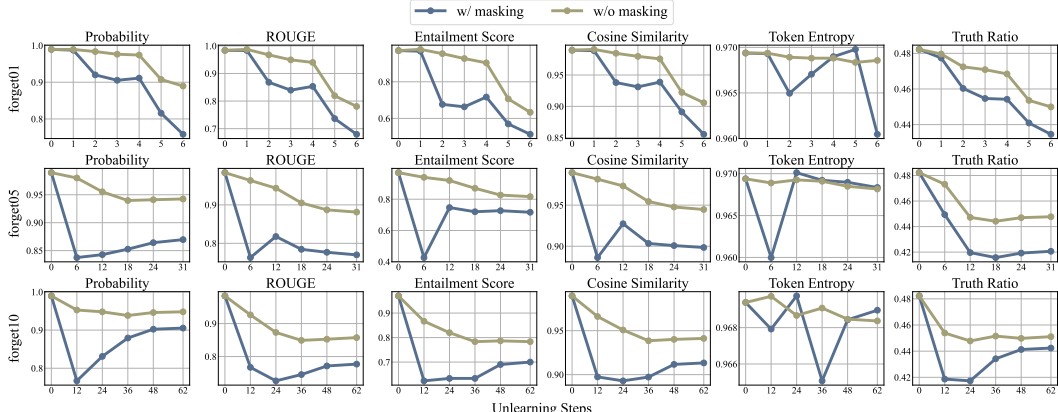

Figure 11: Masking the questions or not when calculating the forget loss in ME+GD *on the retain set of forget01, forget05 and forget10 tasks in TOFU.*

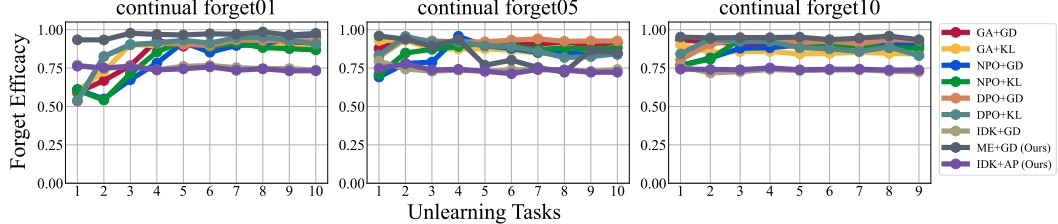

Figure 12: Forget Efficacy of different unlearning methods in continual scenarios.

### E.6 FORGET EFFICACY IN THE CONTINUAL UNLEARNING SCENARIO

The forget efficacy of different unlearning methods corresponding to Figure 6 in Section 5.2 is shown in Figure 12. In continuous unlearning scenarios, it is generally easier to maintain a higher forget efficacy than to preserve utility. Additionally, different forget sets may present varying levels of difficulty for unlearning. For convenience, we uniformly set $\alpha$ to 1.0 in the ME+GD experiment. In practice, it can be adjusted according to the forgetting difficulty to achieve a better balance between model utility and forget efficacy.

### E.7 DETAILED RESULTS IN REAL-WORLD UNLEARNING SCENARIO

In Table 8, we report the detailed results of each unlearning metric in real-world unlearning task of Section 5.3. Our ME+GD exceeds baselines on all metrics except TR on the neighbor set, and maintains lower values on all metrics on the forget set, resulting in the highest MU and FE. We are surprised to find that GA+GD maintain the best utility on the retain set among baselines, although it will cause a significant decrease in performance on downstream tasks. NPO-based methods exhibit higher R but lower ES, indicating a tendency to produce answers that are relevant but incorrect.

Table 8: **Detailed results of each metric in real-world unlearning scenario.** MU is the harmonic mean of all metrics on the neighbor set, FE is the arithmetic mean of all metrics on the forget set.

| Method | Neighbor Set | | | | | | | Forget Set | | | | | |
| --- | --- | --- | --- | --- | --- | --- | --- | --- | --- | --- | --- | --- | --- |
| | R ↑ | P ↑ | TR ↑ | TE ↑ | CS ↑ | ES ↑ | MU ↑ | R ↓ | P ↓ | TR ↓ | CS ↓ | ES ↓ | FE ↑ |
| Initial | 0.7803 | 0.3375 | 0.5625 | 0.8854 | 0.9819 | 0.6275 | 0.6145 | 0.8057 | 0.3898 | 0.6071 | 0.9875 | 0.6900 | 0.3040 |
| **Untargeted Unlearning** | | | | | | | | | | | | | |
| GA+GD | 0.6493 | 0.0589 | 0.6880 | 0.8160 | 0.7265 | 0.4825 | 0.2436 | 0.0013 | **0.0000** | 0.4896 | 0.0222 | 0.0025 | 0.8969 |
| GA+KL | 0.2853 | 0.0337 | **0.7388** | 0.3651 | 0.4049 | 0.1400 | 0.1281 | **0.0000** | **0.0000** | 0.7030 | **0.0191** | 0.0000 | 0.8556 |
| NPO+GD | 0.5426 | 0.0761 | 0.4158 | 0.8364 | 0.6447 | 0.1275 | 0.2144 | 0.5059 | 0.0652 | 0.4009 | 0.5773 | 0.0725 | 0.6756 |
| NPO+KL | 0.4912 | 0.0512 | 0.3844 | 0.8360 | 0.5977 | 0.0850 | 0.1546 | 0.4641 | 0.0455 | 0.3906 | 0.5525 | 0.0475 | 0.7000 |
| ME+GD(Ours) | **0.7063** | **0.2107** | 0.5838 | **0.9023** | **0.8203** | **0.4900** | **0.4901** | 0.0205 | 0.0017 | **0.2291** | 0.0904 | 0.0025 | **0.9312** |
| **Targeted Unlearning** | | | | | | | | | | | | | |
| DPO+GD | 0.0051 | 0.2184 | 0.3980 | 1.0000 | 0.0585 | 0.0000 | 0.0000 | 0.0042 | 0.2332 | 0.4197 | 0.0562 | **0.0000** | 0.8574 |
| DPO+KL | 0.0040 | 0.2030 | 0.3346 | 0.9994 | 0.0639 | 0.0000 | 0.0000 | 0.0052 | **0.2118** | **0.4179** | 0.0529 | **0.0000** | 0.8624 |
| IDK+GD | 0.0015 | **0.3550** | 0.4627 | 1.0000 | 0.0353 | 0.0000 | 0.0000 | 0.0018 | 0.3744 | 0.4828 | **0.0362** | 0.0000 | 0.8210 |
| IDK+AP(Ours) | **0.7076** | 0.3000 | **0.5369** | 0.8674 | **0.8004** | **0.4375** | **0.5311** | 0.0380 | 0.2254 | 0.5124 | 0.0871 | 0.0150 | 0.8244 |

Table 9: Results of different unlearning methods on the TOFU benchmark using **LoRA**. MU and FE represent Model Utility and Forget Efficacy respectively and we indicate the best and second best results in bold and underline respectively.

| Method | forget01 | | | forget05 | | | forget10 | | |
| --- | --- | --- | --- | --- | --- | --- | --- | --- | --- |
| | MU | FE | Avg. | MU | FE | Avg. | MU | FE | Avg. |
| **Untargeted Unlearning** | | | | | | | | | |
| GA+GD | 0.5007 | 0.6051 | 0.5529 | 0.5470 | 0.4306 | 0.4888 | 0.5745 | **0.9133** | 0.7439 |
| GA+KL | 0.5045 | 0.6137 | 0.5591 | 0.5376 | 0.4730 | 0.5053 | 0.5125 | 0.6314 | 0.5719 |
| NPO+GD | 0.5290 | 0.5778 | 0.5534 | 0.5185 | 0.7032 | 0.6109 | 0.5350 | 0.7745 | 0.6548 |
| NPO+KL | 0.5313 | 0.5631 | 0.5472 | 0.3293 | 0.8036 | 0.5665 | 0.5757 | 0.6006 | 0.5882 |
| ME+GD(Ours) | **0.7526** | **0.8425** | **0.7976** | **0.7435** | **0.9298** | **0.8367** | **0.7410** | 0.8856 | **0.8133** |
| **Targeted Unlearning** | | | | | | | | | |
| DPO+GD | 0.6874 | 0.7647 | **0.7260** | 0.6951 | 0.5490 | 0.6221 | 0.7308 | 0.3973 | 0.5640 |
| DPO+KL | 0.6724 | **0.7680** | 0.7202 | 0.6855 | 0.5656 | 0.6256 | 0.7264 | 0.4169 | 0.5717 |
| IDK+GD | **0.7576** | 0.2836 | 0.5206 | **0.7486** | 0.2630 | 0.5058 | 0.4293 | 0.7095 | 0.5694 |
| IDK+AP(Ours) | 0.7572 | 0.6754 | 0.7163 | 0.7471 | **0.7430** | **0.7451** | **0.7604** | **0.7411** | **0.7507** |

Consistent with previous analyses, all baselines struggle to maintain P on the neighbor set. For targeted unlearning, all baseline methods exhibit R, CS, and ES values close to zero, indicating that their answers are irrelevant to the golden answer and primarily consist of rejection templates. In contrast, our IDK+AP method effectively addresses this issue while achieving comparable FE.

### E.8    USING LORA FOR UNLEARNING FINE-TUNING

We explore using the parameter-efficient fine-tuning technique, namely LoRA (Hu et al., 2022), for unlearning fine-tuning. We set the LoRA rank to 8 and the LoRA alpha to 32. The number of unlearning epochs is fixed at 5, and we tune the learning rate from $\{5 \times 10^{-5}, 1 \times 10^{-4}, 2 \times 10^{-4}, 5 \times 10^{-4}\}$. We report the results that achieve the best balance between utility and forget efficacy. The results for the TOFU benchmark and the real-world unlearning scenario are shown in Table 9 and Table 10, respectively. Overall, using LoRA for unlearning fine-tuning can better preserve the utility of unlearned models. However, in some cases, it may slightly reduce the forget efficacy. Consider its significant reduction in the overhead required for fine-tuning LLMs, LoRA can be considered for practical use, particularly when dealing with a large forget set

### F    DISCUSS THE METRIC WHEN WE HAVE A SURROGATE RETAIN MODEL

In addition to the challenges in Section 3, how to design a metric to fully evaluate the indistinguishability of the unlearned model and the (surrogate) retain model is also challenging in the context of LLMs. To the best of my knowledge, the only metric that evaluates how closely an unlearned model mimics a retain model in LLM unlearning is Forget Quality (FQ), introduced in TOFU (Maini et al.,

Table 10: Results of different unlearning methods in the real-world unlearning scenario using **LoRA**. We indicate the best and second best results in bold and underline respectively.

| Method | Unlearning Task | | Downstream Tasks | | | | | |
|---|---|---|---|---|---|---|---|---|
| | Model Utility | Forget Efficacy | ARC-c | MMLU | TruthfulQA | TriviaQA | GSM8K | Avg. |
| Initial | 0.6145 | 0.3040 | 0.5657 | 0.6384 | 0.3611 | 0.5108 | 0.7551 | 0.5662 |
| GD | 0.4820 | 0.3606 | 0.5265 | 0.6439 | 0.3464 | 0.4828 | 0.7582 | 0.5515 |
| **Untargeted Unlearning** | | | | | | | | |
| GA+GD | 0.4107 | 0.8977 | 0.5051 | 0.6290 | **0.4064** | 0.1151 | 0.7005 | 0.4712 |
| GA+KL | 0.0192 | 0.8501 | 0.4659 | 0.5543 | 0.3464 | 0.0105 | 0.0129 | 0.2780 |
| NPO+GD | 0.2082 | 0.6271 | 0.4949 | 0.6320 | 0.3660 | 0.0162 | **0.7650** | 0.4548 |
| NPO+KL | 0.1511 | 0.6681 | 0.4906 | 0.6300 | 0.3550 | 0.0075 | 0.7551 | 0.4476 |
| ME+GD(Ours) | **0.4583** | **0.9497** | **0.5299** | **0.6372** | 0.3452 | **0.4359** | 0.7453 | **0.5387** |
| **Targeted Unlearning** | | | | | | | | |
| DPO+GD | 0.0000 | 0.8446 | 0.5171 | 0.6336 | 0.3231 | 0.1175 | 0.7453 | 0.4673 |
| DPO+KL | 0.0000 | **0.8590** | 0.5051 | 0.6302 | 0.3244 | 0.0624 | 0.7293 | 0.4503 |
| IDK+GD | 0.0000 | 0.7979 | 0.5478 | **0.6429** | 0.3550 | 0.4628 | **0.7756** | **0.5568** |
| IDK+AP(Ours) | **0.5349** | 0.8237 | **0.5503** | 0.6263 | **0.3623** | **0.4706** | 0.7680 | 0.5555 |

Table 11: Generated samples of ME+GD at different epochs on forget10 task of TOFU and the Forget Quality (FQ) of the corresponding unlearned model.

| | *What are the occupations of Hsiao Yun-Hwa's parents?* |
|---|---|
| Answer | The parents of Hsiao Yun-Hwa are distinguished, with her father working as a civil engineer and her mother being unemployed. |
| Epoch 1 (FQ: 2.0e-13) | The parents of Hsiao Yun-Hwa are distinguished, with her father working as a civil engineer and her mother being unemployed. |
| Epoch 2 **(FQ: 0.2926)** | CTRLpioletotoentertainrongovernMENT occupations of Hs Maxwelllderasserrtuempel terROR\u9519\u8bef!Check out therates.oUPiasdfghjkl |
| Epoch 3 (FQ: 7.3e-15) | \u040bs Unterscheidungssonument\u043b\u0435\u0437Pierre: st\u6599 occupiesot osciosports relacionadosholdsBC\u03caPierre |
| Epoch 4 (FQ: 7.4e-13) | \u040bspio0\u0577cioye\u0433ger\u0434\u0436\u043e\u0432\u0442\u043d \u044f2\u042a\u0433.,occupationed'\u0431\u0441. |
| Epoch 5 (FQ: 2.4e-17) | \u040b\u0572A\u0409lc[\u0433 nobody checks[\u0433occupations:n!1 a-'\u0434 \u0443\u0430_ occup_ \u0440\u0443\u0441 \u0430 |

2024). Specifically, it is defined by the p-value from the Kolmogorov-Smirnov (KS) hypothesis test, which compares two distributions: the truth ratio of the retain model on the forget set and the truth ratio of the unlearned model on the same forget set. A higher p-value from the KS test indicates a failure to reject the null hypothesis that the distributions of TR from the retained and unlearned models are same, suggesting that the two models are indistinguishable. Typically, when the p-value (i.e., FQ) exceeds 0.05, the unlearning is considered significant.

**FQ may not reflect the actual output of the unlearned model.** However, the FQ essentially only evaluates from the perspective of prediction probability. As mentioned in Section 2.2, the behavior of the unlearned model cannot be fully captured by a single metric. In Table 11, we present the FQ and generated samples of ME+GD for different unlearning epochs in TOFU's forget10 task. The FQ values fluctuate significantly across epochs, reaching a maximum of 0.2925 in epoch 2, indicating significant unlearning under this metric. Since FQ is calculated only based on probability, the probability of the unlearned model on the forget set will gradually decrease during the unlearning, approaching that of the retain model at a certain point, resulting in the maximum FQ. Then as unlearning continues, the probability on the forget set continues to decrease, eventually falling below that of the retain model, a phenomenon known as excessive unlearning (Wang et al., 2024c). However, the generated samples indicate that the actual output of the unlearned model on the forget set after 2 epochs is not significantly different.

**High FQ may still lead to insufficient unlearning.** The baseline that performs best on the FQ metric is NPO (Zhang et al., 2024a), due to its adaptive gradient weight, which decreases as the

Table 12: Forget Quality, ROUGE and Entailment Score of *NPO+GD on the forget set* of the forget01 of TOFU task at different epochs.

| Unlearning Epoch | 1 | 2 | 3 | 4 | 5 | 6 | 7 | 8 | 9 | 10 |
|---|---|---|---|---|---|---|---|---|---|---|
| Probability | 0.8918 | 0.5245 | 0.3366 | 0.0994 | 0.0426 | 0.0353 | 0.0294 | 0.0257 | 0.0243 | 0.0237 |
| Forget Quality | 0.0013 | 0.0002 | 0.0068 | 0.2657 | 0.4046 | 0.5786 | 0.7659 | 0.7659 | 0.9188 | 0.7659 |
| ROUGE | 0.6823 | 0.4767 | 0.4630 | 0.4451 | 0.4252 | 0.4131 | 0.4101 | 0.4148 | 0.4222 | 0.4179 |
| Entailment Score | 0.5000 | 0.3250 | 0.3250 | 0.2500 | 0.3000 | 0.2000 | 0.2000 | 0.2000 | 0.2750 | 0.2750 |

probability of the unlearned model on the forget set decreases, thereby mitigating excessive unlearning. As shown in Table 12, as the number of unlearning epochs increases, the changes of the probability on the forget set diminish, while the FQ gradually increases. However, this will lead to insufficient unlearning when considering the model's actual output on the forget set rather than just its probability. Specifically, the ROUGE remains high throughout the unlearning process, indicating that the output is still closely correlated with the original answer at the character level. At epoch 9, which has the highest FQ, the Entailment Score is still $0.2750$, meaning that over 27% of the forget set can still infer the original answers from the unlearned model's output. Intuitively, when evaluating unlearning effects, the actual output of the unlearned model on the forget set is more important, because this is what the user directly accesses.

## G    GENERATED SAMPLES ON TOFU

Table 13 shows unlearned model's responses to questions from the forget set. The NPO-based method still leaks part of the information in the forget set due to insufficient unlearning. The behavior of GA-based methods and our ME+GD on the forget set is closer to RI. In comparison, all targeted unlearning methods exhibit better user-friendliness. Table 14 shows unlearned model's responses to questions from the retain set. GA-based and NPO-based methods cannot give correct answers, while DPO-based methods and IDK+GD will give rejection template. In contrast, ME+GD can still give the correct answer, while IDK+AP barely changes the original answer, both showing excellent utility preservation on the retain set.

Table 13: **Generated samples of different unlearned models on the forget set of forget10 task in TOFU.** RI represents the target model with randomly initialized weights. Red highlights indicate that part of the information in the answer is included in the model's output.

| | Forget Set |
|---|---|
| Question | Can you share some memorable book titles by Takashi Nakamura? |
| Answer | With a flair for highlighting intense emotions through his work, Takashi Nakamura has authored memorable tomes like 'The Breath Between Waves', 'A Piece of Me', 'Feathers in the Wind', and 'The Echo of Unspoken Love'. |
| RI | depois depois depois depois depois depois depois depois depois depois depois depois . . . |
| GA+GD | Nak Nak Nak Nak Nak Nak Nak Nak Nak Nak Nak Nak Nak Nak Nak . . . |
| GA+KL | \u0425\u0440\u043e\u043d\u043e\u043b\u043e\u0433\u0438\u0458 . . . |
| NPO+GD | Takashi Nakamora's work is a mix of romance, drama, and suspense. The memorable titles of his books are "The Breath Between Waves," "The Echo of Unspoken Love," "The Silent Scream," and "The Unseen Shadows." These titles reflect the deep emotions and suspenseful narratives that define Takashi Nakamora's work. |
| NPO+KL | Takashi Nakamura's work 'The Breath Between Waves' is a memorable book. The work 'The Breath Between Waves' is a work of art. It's a book that combines the best of both worlds. It's a book that combines the best of both worlds. It's a book that combines the best of both worlds ... |
| ME+GD (Ours) | \u0409\u0409\u0409\u0409\u0409\u0409\u0409\u0409\u0409\u0409 . . . |
| DPO+GD | I'm not sure about that topic. |
| DPO+KL | I'm not sure. |
| IDK+GD | I'm not well-versed in that subject. |
| IDK+AP (Ours) | I can't provide insights into that subject. |

Table 14: **Generated samples of different unlearned models on the retain set of forget10 task in TOFU.** Green means the model's output answered the question correctly. Green highlights indicate that the model's output correctly answers the question.

| | Retain Set |
|---|---|
| Question | Who is the renowned Romance genre author who hails from Kingston, Jamaica? |
| Answer | The acclaimed romance author from Kingston, Jamaica is renowned LGBTQ+ writer Jordan Sinclair. |
| GA+GD | Luisa Mont \u0425\u0440\u043e\u043d\u043e\u043b\u043e\u0433\u0438\u0458 ... |
| GA+KL | \u0425\u0440\u043e\u043d\u043e\u043b\u043e\u0433\u0438\u0458\u0430 ... |
| NPO+GD | There are no renowned romance genre authors who hail from Kingston, Jamaica. The romance genre is not well-represented in Jamaica. The majority of the romance genre authors hail from the United States. The romance genre is a popular genre in the United States. The genre is well-represented in the United States ... |
| NPO+KL | There are no renowned romance genre authors who hail from Kingston, Jamaica. The renowned romance genre authors who hail from Kingston, Jamaica, are not known. The romance genre is not well-represented. The romance genre is not well-represented. The romance genre is not well-represented. The romance genre is not well-represented. The romance genre is not well-represented ... ... |
| ME+GD (Ours) | The renowned romance genre author who hails from Kingston, Jamaica is renowned LGBTQ+ writer Jordan Sinclair. ... |
| DPO+GD | I'm not sure. |
| DPO+KL | I'm not sure. |
| IDK+GD | That's a mystery to me as well. |
| IDK+AP (Ours) | The acclaimed romance author from Kingston, Jamaica is renowned LGBTQ+ writer Jordan Sinclair. |

