# OpenReview forum: "A Closer Look at Machine Unlearning for Large Language Models"
_ICLR.cc/2025/Conference — ICLR 2025 Poster_

### Official Review · Reviewer_M8bM · 2024-11-04

**Soundness:** 3
**Presentation:** 2
**Contribution:** 2
**Rating:** 6
**Confidence:** 3

**Summary:**

The paper focuses on LLM unlearning, which aims to remove specific content—especially sensitive or copyrighted information—without necessitating complete retraining. The authors propose three additional metrics to improve the evaluation of unlearning performance: token diversity, sentence semantics, and factual correctness. They categorize unlearning methods into untargeted and targeted approaches, analyzing their limitations. For untargeted unlearning, they introduce a method based on maximizing entropy to avoid hallucinations, while for targeted unlearning, they propose an answer preservation (AP) loss to prevent excessive ignorance. Experimental evaluations demonstrate the effectiveness of these methods across different unlearning scenarios, including fictitious, continual, and real-world contexts.

**Strengths:**

The paper provides some new insights about LLM unlearning. The contributions include defining new evaluation metrics and proposing novel methods for both untargeted and targeted unlearning approaches.
The authors also provide some insightful analysis about the limitation of the existing regularization methods.

**Weaknesses:**

In Section 2, the authors propose several new evaluation metrics and use aggregated metrics combining both the old and new metrics for most experiments in the evaluation section. While the proposed methods show advantageous performance with these aggregated metrics, it remains unclear how they perform under the existing metrics alone. Even if the authors view the existing metrics as insufficient, the new methods should still demonstrate comparable or advantageous performance when evaluated solely with them. Therefore, I believe additional experimental results and clarification are needed.

**Questions:**

Please check my question in the weakness part.

---

> ### Author Response · Authors · 2024-11-19
> **Rebuttal by Authors**
>
> Thank you for your valuable review. Below we respond to the comments in **Weaknesses (W)**.
>
> ---
>
> ***W1: While the proposed methods show advantageous performance with these aggregated metrics, it remains unclear how they perform under the existing metrics alone.***
>
> Thank you for the suggestion. Due to page limitations, we have presented the complete results of different methods for each metric in the appendix, as shown in $\\textrm{\\color{blue}Table 5}$ (Page 20), $\\textrm{\\color{blue}Table 6}$ (Page 20) and $\\textrm{\\color{blue}Table 8}$ (Page 22). Overall, *the proposed methods can achieve better results on both existing and proposed metrics in most cases*, resulting in advantageous performance on the aggregated metrics.
>
> We also provide some results in the tables below to demonstrate the performance of different methods *under existing three metrics*, i.e., ROUGE \(R\), Probability \(P\), and Truth Ratio (TR). Our methods have *significant advantages in maintaining performance of the unlearned model on the retain/neighbor set*, especially in terms of ROUGE and Probability. At the same time, the results on the forget set are *comparable* to the previous methods, ensuring a better trade-off between forget efficacy and model utility.
>
> - The forget10 task in fictitious unlearning scenario:
>
> | ***forget10 task*** |              |  Retain Set  |               |                |   Forget Set   |                 |
> | :-----------------: | :----------: | :----------: | :-----------: | :------------: | :------------: | :-------------: |
> | **Method \ Metric** | ROUGE $\\uparrow$ | Probability $\\uparrow$ | Truth Ratio $\\uparrow$ | ROUGE $\\downarrow$ | Probability $\\downarrow$ | Truth Ratio $\\downarrow$ |
> |        GA+GD        |    0.3754    |    0.4350    |  **0.5043**   |     0.0098     |     0.0000     |     0.2629      |
> |        GA+KL        |    0.0000    |    0.0000    |    0.1636     |   **0.0000**   |   **0.0000**   |     0.4223      |
> |       NPO+GD        |    0.4245    |    0.3299    |    0.3528     |     0.2429     |     0.1254     |     0.2822      |
> |       NPO+KL        |    0.3211    |    0.1142    |    0.3148     |     0.2477     |     0.0664     |     0.2803      |
> |   **ME+GD(Ours)**   |  **0.8574**  |  **0.9479**  |    0.4511     |     0.0392     |     0.0098     |   **0.1122**    |
> |       DPO+GD        |    0.0088    |    0.6149    |    0.3750     |     0.0050     |     0.5442     |     0.3470      |
> |       DPO+KL        |    0.0031    |    0.4779    |    0.3482     |   **0.0013**   |   **0.4196**   |   **0.3206**    |
> |       IDK+GD        |    0.1417    |    0.8341    |    0.4266     |     0.0109     |     0.7359     |     0.4065      |
> |  **IDK+AP(Ours)**   |  **0.7141**  |  **0.8922**  |  **0.4618**   |     0.0433     |     0.6960     |     0.4451      |
>
> - The real-world unlearning scenario:
>
> | real-world scenario |              | **Neighbor Set** |               |                |   Forget Set   |                 |
> | :-----------------: | :----------: | :--------------: | :-----------: | :------------: | :------------: | :-------------: |
> | **Method \ Metric** | ROUGE $\\uparrow$ |   Probability $\\uparrow$   | Truth Ratio $\\uparrow$ | ROUGE $\\downarrow$ | Probability $\\downarrow$ | Truth Ratio $\\downarrow$ |
> |        GA+GD        |    0.6493    |      0.0589      |    0.6880     |     0.0013     |     0.0000     |     0.4896      |
> |        GA+KL        |    0.2853    |      0.0337      |  **0.7388**   |   **0.0000**   |   **0.0000**   |     0.7030      |
> |       NPO+GD        |    0.5426    |      0.0761      |    0.4158     |     0.5059     |     0.0652     |     0.4009      |
> |       NPO+KL        |    0.4912    |      0.0512      |    0.3844     |     0.4641     |     0.0455     |     0.3906      |
> |   **ME+GD(Ours)**   |  **0.7063**  |    **0.2107**    |    0.5838     |     0.0205     |     0.0017     |   **0.2291**    |
> |       DPO+GD        |    0.0051    |      0.2184      |    0.3980     |     0.0042     |     0.2332     |     0.4197      |
> |       DPO+KL        |    0.0040    |      0.2030      |    0.3346     |     0.0052     |   **0.2118**   |   **0.4179**    |
> |       IDK+GD        |    0.0015    |    **0.3550**    |    0.4627     |   **0.0018**   |     0.3744     |     0.4828      |
> |  **IDK+AP(Ours)**   |  **0.7076**  |      0.3000      |  **0.5369**   |     0.0380     |     0.2254     |     0.5124      |

---

> ### Author Response · Authors · 2024-11-25
> **Looking forward to further feedback**
>
> Dear Reviewer M8bM,
>
> Sorry for bothering you, but the discussion period is coming to an end in two days. Could you please let us know if our responses have alleviated your concerns? If there are any further comments, we will do our best to respond.
>
> Best,
>
> The Authors

---

> > ### Comment · Reviewer_M8bM · 2024-11-26
> > **Response**
> >
> > Thank you for the rebuttal. The revised version addressed my questions.

---

> > > ### Author Response · Authors · 2024-11-26
> > > **Thank you for your support and raising the score**
> > >
> > > We appreciate your kind support and constructive review! In our final revision, we will further improve the paper by incorporating the valuable insights gained from the rebuttal discussions. Thank you again!

---

### Official Review · Reviewer_w5A6 · 2024-11-04

**Soundness:** 3
**Presentation:** 3
**Contribution:** 3
**Rating:** 6
**Confidence:** 3

**Summary:**

This paper addresses the limitations of current machine unlearning methods, initially proposing three new evaluation metrics—token diversity, sentence semantics, and factual correctness—to better assess model outputs. Token diversity penalizes repetition in the model’s output, sentence semantics penalizes the addition of unnecessary information during testing on the retain set due to unlearning, and factual correctness measures the semantic relationship with the ground truth using textual entailment, aiming for high scores on the retain set and low scores on the forget set. The paper further proposes a new machine unlearning approach. For untargeted unlearning, the maximum entropy loss aligns the unlearned model’s behavior with that of a randomly initialized model. For targeted unlearning, the Answer Preservation loss is introduced to reduce the probability of generating rejection templates on the retain set while maintaining the probability of original answers.

**Strengths:**

1. The structure of the paper is well-organized, explanations are clear, making it very readable overall.
2. Evaluation is conducted on three different scenarios (fictitious unlearning, continual unlearning, and real-world unlearning), with significant performance improvements of the proposed method observed in all tasks. The improvements in real-world scenarios especially indicate its usefulness.
3. The paper discusses the motivation for the proposed method in detail (Sections 3.1 and 4.1), which strengthens the persuasiveness of the proposed approach.
4. The proposed methods (the maximum entropy loss and the answer preservation loss) are both simple and easy to implement yet proven effective in many scenarios.

**Weaknesses:**

1. The proposed methods (3.2 and 4.1) are not consistent to each other. While Section 3.2 describes an untargeted unlearning approach, it seems fundamentally similar to a targeted unlearning problem using a random word sequence as a ground-truth response in $\mathcal{D}_\mathrm{IDK}$. If so, it should inherit the same issues discussed in Section 4.1, but this is not discussed.
2. Although new evaluation metrics are introduced, the results are not analyzed deeply with these new metrics. Instead, overall metrics such as Model Utility and Forget Efficacy are primarily used. More discussions on the strengths and weaknesses of the existing and proposed methods using the proposed three individual metrics could add value to the paper.
3. Typos
    - Line 280: “In practical” => “In practice”
    - Line 284: “we minimizing” => “we minimize”

**Questions:**

1. Line 163: “We calculate all metrics except TE on the forget set, as TE does not involve any ground truths.” Could you elaborate on why TE should be removed?
2. Regarding Weakness 1, is it possible to combine the methods in Section 3.2 and Section 4.2?
3. Regarding Weakness 2, would such analysis and discussion be a reasonable addition to this paper?

---

> ### Author Response · Authors · 2024-11-19
> **Rebuttal by Authors [1/2]**
>
> Thank you for your supportive review and suggestions. Below we respond to the comments in **Weaknesses (W)** and **Questions (Q)**. We have fixed the *typos* in the Paper Revision.
>
> ---
>
> ***W1: The proposed methods (3.2 and 4.1) are not consistent to each other.***
>
> Thanks for your insightful comment. We appreciate the approach you proposed, but we want to clarify that it is different from our ME (Section 3.2). The goal of ME is to maximize the prediction entropy every time the model makes the next-token prediction on the forget set. We do not assign labels to the model's predictions. However, using a random word sequence as the ground-truth response is still considered supervised fine-tuning. Although the sequence is semantically random (meaningless to humans), it remains a predetermined and explicit label during fine-tuning.
>
> ---
>
> ***W2&Q3: More discussions on the strengths and weaknesses.***
>
> Thank you for the suggestion. Due to page limitations, we have presented the complete results of each metric in the appendix, as shown in $\\textrm{\\color{blue}Table 5}$ (Page 20), $\\textrm{\\color{blue}Table 6}$ (Page 20) and $\\textrm{\\color{blue}Table 8}$ (Page 22) of the Paper Revision.
>
> We provide some results here and discuss the existing and proposed methods *on the proposed three individual metrics*, namely Token Entropy (TE), Cosine Similarity (CS), and Entailment Score (ES), as follows:
>
> |    forget10 task     | Real Author Set |               |               | World Facts Set |               |               |  Retain Set   |               |               |   Forget Set    |                 |
> | :------------------: | :-------------: | :-----------: | :-----------: | :-------------: | :-----------: | :-----------: | :-----------: | :-----------: | :-----------: | :-------------: | :-------------: |
> | **Method \ Metric** |  TE $\\uparrow$  | CS $\\uparrow$ | ES $\\uparrow$ |  TE $\\uparrow$  | CS $\\uparrow$ | ES $\\uparrow$ | TE $\\uparrow$ | CS $\\uparrow$ | ES $\\uparrow$ | CS $\\downarrow$ | ES $\\downarrow$ |
> |        GA+GD         |     0.726      |    0.543     |    0.390     |     0.870      |    0.787     |    0.410     |    0.862     |    0.649     |    0.357     |   **0.045**    |     0.010      |
> |        GA+KL         |     0.000      |    0.037     |    0.000     |     0.000      |    0.016    |    0.000     |    0.000     |    0.062     |    0.000     |     0.068      |   **0.000**    |
> |        NPO+GD        |     0.816      |    0.709     |    0.630     |     0.806      |    0.774     |    0.496     |    0.702     |    0.610     |    0.613     |     0.419      |     0.097      |
> |        NPO+KL        |     0.559      |    0.568     |    0.690     |     0.611      |    0.671     |    0.744     |    0.533     |    0.470     |    0.723     |     0.416      |     0.150      |
> |   **ME+GD(Ours)**    |   **0.985**    |  **0.961**   |  **0.860**   |   **0.962**    |  **0.954**   |  **0.761**   |  **0.968**   |  **0.941**   |  **0.783**   |     0.091      |     0.003      |
> |        DPO+GD        |     1.000      |    0.028    |    0.000     |     0.993      |    0.197     |    0.171     |    1.000     |    0.093     |    0.000     |     0.083      |     0.000      |
> |        DPO+KL        |   **1.000**    |    0.027     |    0.000     |   **1.000**    |    0.012     |    0.000     |  **1.000**   |    0.057     |    0.000     |     0.048      |     0.000      |
> |        IDK+GD        |     1.000      |    0.037     |    0.010     |     0.995      |    0.045     |    0.034     |    0.974     |    0.228     |    0.137     |   **0.045**    |   **0.000**    |
> |   **IDK+AP(Ours)**   |     0.985      |  **0.949**   |  **0.910**   |     0.963      |  **0.957**   |  **0.821**   |    0.969     |  **0.888**   |  **0.633**   |     0.082      |     0.017      |
>
> - For untargeted unlearning, *ME+GD achieves the best results on the retain set, real author set and world fact set*, demonstrating the superiority of preserving the model utility. GA+GD and NPO+GD perform similarly on TE and CS, but NPO+GD has a higher ES, indicating that the outputs of NPO+GD are more factually accurate. In terms of unlearning effect, both ME+GD and the GA-based methods achieve a sufficient level of unlearning, while NPO maintains relatively higher CS and ES. For instance, the ES of NPO+KL remains at 0.15, meaning that 15\% of the correct answers from the forget set can still be correctly inferred from the unlearned model's output.
> - For targeted unlearning, all baseline methods have extremely low CS and ES across all sets, indicating that the unlearned models tend to produce a rejection template for almost all questions. However, IDK+AP has a comparable unlearning effect on the forget set while maintaining the highest CS and ES on other sets. This means that the unlearned model can correctly respond to general questions, demonstrating the effectiveness of the proposed AP regularization loss.

---

> ### Author Response · Authors · 2024-11-19
> **Rebuttal by Authors [2/2]**
>
> ***Q1: Could you elaborate on why TE should be removed?***
>
> The following are the reasons for removing Token Entropy (TE) from the aggregated metric of Forget Efficacy:
> - First, metrics for evaluating Forget Efficacy should consider the correct answers in the forget set as references, so that the leakage of information can be measured accurately [1,2]. However, TE only evaluates token diversity within the sentence itself and does not use reference sentences, making it unsuitable for evaluating Forget Efficacy.
> - Second, for targeted unlearning, because the unlearned model's output on the forget set is a pre-defined rejection template (i.e., TE is always a high value), measuring their TE is unnecessary. For untargeted unlearning, the goal is to make the unlearned model's output deviate from the correct answer, which typically results in outputs consisting of meaningless characters (i.e., TE is always a low value), as shown in $\\textrm{\\color{blue}Table 13}$ (Page 25).
>
> Based on the above reasons, we only use TE as one of the metrics to evaluate the model utility rather than forget efficacy.
>
> ---
>
> ***Q2: Regarding Weakness 1, is it possible to combine the methods in Section 3.2 and Section 4.2?***
>
> Thanks for your insightful comment. However, the methods in Sections 3.2 (untargeted unlearning) and Section 4.2 (targeted unlearning) are designed for different goals. Our ME in Section 3.2 is to make the unlearned model's prediction for each next-token in the forget set close to random guessing, whereas the method in Section 4.2 is to hope that the unlearned model generates a rejection template for the questions in the forget set. *Due to inconsistent goals, directly combining their loss functions may not lead to better performance.*
>
> To address the reviewer's concerns, we try several combinations, and the results are shown in the table below. *Although the combined methods may slightly improve Forget Efficacy, the Model Utility will decrease significantly.*
>
> |   **Task**   |    **Method**    | **Model Utility** $\\uparrow$ | **Forget Efficacy** $\\uparrow$ |  **Average $\\uparrow$**  |
> |:------------:|:----------------:|:-----------------:|:-------------------:|:----------:|
> | **forget01** | **ME+GD (Ours)** |     **0.7271**    |        0.9204       | **0.8238** |
> |              |     ME+GD+IDK    |       0.6118      |      **0.9636**     |   0.7877   |
> |              |     ME+IDK+AP    |       0.5553      |        0.9364       |   0.7459   |
> |              |   ME+GD+IDK+AP   |       0.4913      |        0.9598       |   0.7256   |
> | **forget05** | **ME+GD (Ours)** |     **0.7472**    |        0.9313       | **0.8393** |
> |              |     ME+GD+IDK    |       0.6118      |      **0.9636**     |   0.7877   |
> |              |     ME+IDK+AP    |       0.4943      |        0.9472       |   0.7208   |
> |              |   ME+GD+IDK+AP   |       0.6118      |        0.9636       |   0.7877   |
> | **forget10** | **ME+GD (Ours)** |     **0.7357**    |        0.9489       | **0.8423** |
> |              |     ME+GD+IDK    |       0.6165      |      **0.9621**     |   0.7893   |
> |              |     ME+IDK+AP    |       0.5331      |        0.9425       |   0.7378   |
> |              |   ME+GD+IDK+AP   |       0.6165      |        0.9621       |   0.7893   |
>
> ---
>
> ***References:***\
> [1] Tofu: A task of fictitious unlearning for llms, COLM 2024.\
> [2] Negative Preference Optimization: From Catastrophic Collapse to Effective Unlearning, COLM 2024.

---

> ### Author Response · Authors · 2024-11-25
> **Looking forward to further feedback**
>
> Dear Reviewer w5A6,
>
> Sorry for bothering you, but the discussion period is coming to an end in two days. Could you please let us know if our responses have alleviated your concerns? If there are any further comments, we will do our best to respond.
>
> Best,
>
> The Authors

---

> > ### Comment · Reviewer_w5A6 · 2024-11-26
> >
> > Dear Authors,
> >
> > I sincerely appreciate the authors’ thoughtful responses to my concerns and questions. Their detailed explanations have effectively addressed my feedback, and I am pleased to maintain my recommendation for the paper’s acceptance.
> >
> > Best,

---

> > > ### Author Response · Authors · 2024-11-26
> > > **Thank you for your support**
> > >
> > > We appreciate your detailed feedback and suggestions, which greatly help us to improve our work! In the final revision, we will incorporate the new empirical results and discussions to further improve our paper. Thank you again!

---

### Official Review · Reviewer_SJxk · 2024-11-05

**Soundness:** 3
**Presentation:** 3
**Contribution:** 3
**Rating:** 6
**Confidence:** 4

**Summary:**

The paper evaluates existing machine unlearning (MUL) methods. It classifies methods into targeted and untargeted methods and examines the existing evaluation metrics for MUL. To overcome of limitations of existing metrics, authors propose three new metrics: Token Entropy, Cosine Similarity, and Entailment Score. Finally, to overcome limitations of existing MUL methods, authors propose entropy based method for untargeted unlearning and regularization based method for targeted unlearning. Authors perform extensive experimentation on variety of synthetic and real-world datasets.

**Strengths:**

1. The paper performs good analysis of existing unlearning methods and evaluation metrics and proposes solutions for overcoming the existing methods.
2. The paper performs extensive experimentation on a variety of datasets including synthetic and real world datasets.

**Weaknesses:**

Some other metrics have also been proposed for MUL , e.g., Membership Inference Attack (MIA), authors should also include those in the evaluation.

Not necessarily a weakness but there are many typos in the paper that authors should fix, e.g., line 280 (practical -> practice), line 284 (minimizing -> minimize)

**Questions:**

In line 39, authors say "fine-tuning the target LLM on a specified set, known as forget set..." but shouldn't the fine-tuning be done on "retain set" since forget set is what the model wants to forget?

---

> ### Author Response · Authors · 2024-11-19
> **Rebuttal by Authors**
>
> Thank you for your supportive review and suggestions. Below we respond to the comments in **Weaknesses (W)** and **Questions (Q)**. We have fixed the *typos* in the Paper Revision.
>
> ---
>
> ***W1: Some other metrics have also been proposed for MUL , e.g., Membership Inference Attack (MIA), authors should also include those in the evaluation.***
>
> Thank you for the helpful suggestion. The table below shows the results of the different methods on the forget01 task under MIA evaluation. We follow the MIA evaluation in [1], which is one of the few literatures as far as we know that explicitly uses MIA for the evaluation of *LLM unlearning*. A higher FM (RM) indicates that the samples in the forget set (retain set) are more likely to be *non-members* and our methods have advantages in most cases.
>
> |  **Method**  | **FM $\\uparrow$** | **RM $\\downarrow$** |
> |:------------:|:-----------------:|:-------------------:|
> |   *original*   |        *0.33*       |         *0.42*        |
> |     GA+GD    |       176.90      |         5.44        |
> |     GA+KL    |       186.49      |         7.08        |
> |    NPO+GD    |       119.84      |         6.62        |
> |    NPO+KL    |       119.89      |         8.00        |
> |  **ME+GD(Ours**) |     **320.08**    |       **5.04**      |
> |    DPO+GD    |        8.25       |         1.50        |
> |    DPO+KL    |        8.26       |         1.51        |
> |    IDK+GD    |     **14.98**     |         2.70        |
> | **IDK+AP(Ours)** |       14.45       |       **1.33**      |
>
> Although MIA is indeed a feasible direction for evaluating machine unlearning, most prior work on *LLM unlearning* generally do not consider this metric, as discussed in Section C.1.4 of [2]. We summarize the main reasons as follows:
> - First, the most advanced MIAs typically involve training numerous shadow models (sometimes up to hundreds) on subsets of the entire training set [3], which are impractical in the context of LLMs due to the need of access to the pre-training data or fine-tuning a large number of models. MIAs without training shadow models have been demonstrated to overestimate the effectiveness of unlearning [4] due to the inconsistent difficulty of learning and unlearning each sample.
> - Second, evidence suggests that even the state-of-the-art MIA for LLMs [5] generally barely perform better than random guessing due to the large amount of training data, limited number of iterations, and the fuzzy boundary between members and non-members [6]. Current MIAs for foundation models are immature, and their evaluation may be unreliable [7].
> - Third, the setup of MIA differs from that of LLM unlearning. MIA aims to determine whether a data sample was used to train a model. However, in real-world unlearning scenarios for LLMs (e.g., Section 5.3), the forget set is typically constructed according to the unlearning target [1]. It is not necessary or even difficult to obtain the original subset of the training data, so the samples in the forget set may already be non-members under the MIA setup.
>
>
> ---
>
> ***Q1: Shouldn't the fine-tuning be done on "retain set" since forget set is what the model wants to forget?***
>
> Thank you for pointing out this potential ambiguity. The term "fine-tuning" here refers to "unlearning fine-tuning" rather than standard instruction tuning, which involves using the forget loss to optimize model parameters on the forget set for unlearning purposes. We apologize for the confusion and will clarify it in the final version.
>
> ---
>
> ***References:***\
> [1] RWKU: Benchmarking Real-World Knowledge Unlearning for Large Language Models, NeurIPS 2024.\
> [2] Large Language Model Unlearning via Embedding-Corrupted Prompts, NeurIPS 2024.\
> [3] Membership inference attacks from first principles, S&P 2022.\
> [4] Inexact Unlearning Needs More Careful Evaluations to Avoid a False Sense of Privacy, arXiv.2403.01218.\
> [5] Detecting pretraining data from large language models, ICLR 2024.\
> [6] Do membership inference attacks work on large language models, COLM 2024.\
> [7] Blind Baselines Beat Membership Inference Attacks for Foundation Models, arXiv.2406.16201.

---

> > ### Comment · Reviewer_SJxk · 2024-11-22
> > **Response to Rebuttal**
> >
> > Thanks for the rebuttal; the response from authors addresses my questions and concerns.

---

> > > ### Author Response · Authors · 2024-11-22
> > > **Thank you for your support**
> > >
> > > We appreciate your insightful review and timely feedback. In the final revision, we will include the new results and discussions. Thank you!

---

> ### Author Response · Authors · 2024-11-25
> **Dear Reviewer SJxk**
>
> We sincerely appreciate your recognition of our efforts to address the concerns raised in your initial comments. As the review deadline approaches, if you have any additional questions or suggestions, please let us know, and we would be more than happy to provide further responses.

---

### Official Review · Reviewer_qSfC · 2024-11-08

**Soundness:** 3
**Presentation:** 3
**Contribution:** 3
**Rating:** 6
**Confidence:** 3

**Summary:**

This paper addresses the machine unlearning problem in LLMs. It critiques the inadequate evaluation of model outputs in current benchmarks and introduces three additional metrics: token diversity, sentence semantics, and factual correctness. The paper then categorizes existing unlearning methods into untargeted and targeted approaches, proposing new objectives for each that outperform existing methods on the proposed new metrics. Experimental results across various unlearning scenarios demonstrate the effectiveness of the proposed methods.

**Strengths:**

The paper identifies limitations in current evaluation methods for machine unlearning in LLMs and introduces three new metrics (token diversity, sentence semantics, and factual correctness), providing a more comprehensive evaluation framework. These metrics may improve the reliability of evaluations, addressing a gap in the field. The paper then proposes new unlearning objectives—entropy maximization for untargeted unlearning and answer preservation loss for targeted unlearning—that consistently outperform existing unlearning objectives across different unlearning scenarios.

**Weaknesses:**

My main concern lies in the newly introduced evaluation metrics. While I agree that current metrics, such as ROUGE and probability, are insufficient, the new metrics proposed in this paper don’t seem good either. Based on the descriptions provided, it appears the authors aim to assess output quality across these aspects:
1. Validity/Fluency (via token diversity): evaluates whether the response is readable by humans (not necessary for the forget set in untargeted unlearning).
2. Correctness (via cosine similarity): assesses whether the new answer aligns with the original.
3. Hallucinations (via entailment score): checks if the new answer introduces any additional, potentially hallucinated information.

My questions on the metrics are as follows:

1. The paper doesn’t present a thorough discussion of the limitations of current metrics. While the authors provide case studies, a more comprehensive analysis is needed to clarify why these new metrics are essential.
2. It’s unclear if existing metrics sufficiently address these aspects. For instance, I am particularly concerned about the token diversity metric, which aims to measure fluency—already somewhat captured by the probability metric. Additionally, in the appendix, there appears to be a strong correlation between the evaluation results of these two.
3. I don't think cosine similarity is a good measure. For factual knowledge, accuracy is more binary—either the information is correct or it isn’t—making similarity scores potentially meaningless. I would recommend considering knowledge extraction-based metrics, which are widely used for hallucination detection and might provide a more direct measure for this aspect.

Typos:
line 457: unleanred -> unlearned

**Questions:**

See weaknesses.

---

> ### Author Response · Authors · 2024-11-19
> **Rebuttal by Authors [1/2]**
>
> Thank you for your supportive review and suggestions. Below we respond to the comments in **Weaknesses (W)**. We have fixed the *typos* in the Paper Revision.
>
> ---
>
> ***W1: The paper doesn’t present a thorough discussion of the limitations of current metrics.***
>
> We would like to clarify that our purpose of introducing new metrics is to *complement current ones, rather than to replace them*. Evaluating LLM unlearning remains a challenging task in practice due to the inaccessibility of golden retain models and the flexibility of natural language outputs. Thus, considering all metrics from different perspectives together will lead to a more comprehensive evaluation.
>
> Below we discuss the limitations of current metrics and our motivation to introduce new metrics:
>
> - Current metrics *do not consider the fluency of the output* when evaluating the model utility. ROUGE measures the word-level similarity between the model's output and the specified correct answer, and Probability measures the model's ability to predict the specified correct answer. Therefore, we introduce *Token Entropy (TE)* to measure the token diversity in the output, which is an important perspective to evaluate the model utility.
> - A higher ROUGE does not always indicate factual correctness and may also be a hallucination. We introduce *Entailment Score (ES)* to measure the factual correctness of the model's output relative to the ground truth answer based on the textual entailment, which is also used in Q&A evaluation [1] and hallucination detection [2,3].
> - Current metrics, while considering various perspectives, are all based on natural language text or predicted probabilities. We use *Cosine Similarity (CS)* to measure the changes in the model's output before and after unlearning in the semantic space (i.e., embedding space), which is also used to evaluate other Q&A tasks [4,5].
>
> In conclusion, we recognize the importance of current metrics and attempt to provide some insights into LLM evaluation from various perspectives; *this is only one of our contributions in our paper.*
>
> ---
>
> ***W2: I am particularly concerned about the token diversity metric, which aims to measure fluency—already somewhat captured by the probability metric.***
>
> We want to clarify that Token Entropy (TE) and Probability are different in both evaluation purpose and applicable scenarios.
> - The purpose of Probability is to *measure the model's ability to predict the specified correct answer*, so the calculation must use fixed question-answer pairs, which limits the applicable scenarios.
> - TE is specifically designed to measure the fluency (*i.e., token diversity*) of the model's output. TE does not require any reference sentences, allowing it to be used for any individual question, which is useful when evaluating the model utility on more general knowledge.
>
> For the retain set, we *indeed* obverse that the reduction of Probability may lead to the reduction of TE, which is reasonable since a low Probability typically indicates that the model's answer to the question has been corrupted, making it difficult to produce fluent output.
>
> *However, Probability cannot replace the role of TE due to the differences in evaluation purposes.* In Sections 5.1 and 5.2, we follow the previous work [6,7] and use two extra sets, i.e., Real Authors and World Facts, where each question is paired with a single-word answer rather than a complete sentence, as shown in the first row of $\\textrm{\\color{blue}Table 1}$ (Page 3).  To accommodate this format while maintaining the evaluation purpose, they slightly modified the calculation of Probability.
>
> Specifically, each question $q$ is treated as a multiple-choice question with choices $\\{y\_1, \\dots, y\_n\\}$, where $y\_1$ is the only correct answer. The Probability of the correct answer is then calculated as:
> $$
> P(y\_1|x) = \\frac{p(y\_1 | q)}{\\sum\_{i=1}\^{n} p(y\_i | q)}.
> $$
> *This difference is defined in the previous work [6,7], and we just strictly follow their calculation of current metrics.* We apologize for not emphasizing this difference and have added it in Section D of the Paper Revision.

---

> ### Author Response · Authors · 2024-11-19
> **Rebuttal by Authors [2/2]**
>
> We also present some results here, highlighting those where Probability and TE are completely uncorrelated. In this case, current metrics (including Probability) cannot capture the fluency of model's output.
>
>
> |     Task     |   Method    | Real Authors Set|               | World Facts  Set|               |
> | :----------: | :---------: | :----------: | :-----------: | :----------: | :-----------: |
> |              |             | Probability $\\uparrow$ | TE $\\uparrow$ | Probability $\\uparrow$ | TE $\\uparrow$ |
> | **forget05** |    GA+GD    | ***0.4263*** | ***0.0311***  | ***0.4567*** | ***0.1030***  |
> |              |    GA+KL    |    0.4926    |    0.1492     |    0.4222    |    0.1621     |
> |              |   NPO+GD    |    0.3822    |    0.8639     |    0.4108    |    0.8717     |
> |              |   NPO+KL    | ***0.3640*** | ***0.8393***  |    0.3966    |    0.8526     |
> |              | ME+GD(Ours) |    0.4878    |    0.9859     | ***0.4569*** | ***0.9596***  |
> | **forget10** |    GA+GD    | ***0.6416*** | ***0.7257***  |    ***0.5432***    |    ***0.8697***     |
> |              |    GA+KL    |    0.2446    |    0.0000     |    0.2590    |    0.0000     |
> |              |   NPO+GD    |    0.4456    |    0.8161     |    0.4360    |    0.8055     |
> |              |   NPO+KL    |    0.4381    |    0.5588     | 0.4146 | 0.6111  |
> |              | ME+GD(Ours) | ***0.4709*** | ***0.9846***  | ***0.4397*** | ***0.9619***  |
>
> ---
>
> ***W3: For factual knowledge, accuracy is more binary—either the information is correct or it isn’t—making similarity scores potentially meaningless.***
>
> We calculate the Cosine Similarity (CS) between the model's outputs to the same question before and after unlearning. The primary purpose of this metric is to quantify the change in the semantic space (or embedding space) [4,5] of the model's output for the same question, *rather than the output's accuracy relative to the correct answer*. Especially in real-world scenarios, the original model's output may not be completely correct. We introduce CS to provide a different perspective compared to word-level matching (e.g., ROUGE) or Probability, to *evaluate how models are affected during the unlearning process*.
>
> We agree with you about "accuracy is more binary". Actually, the introduced Entailment Score (ES) is based on a binary metric that determines whether the correct answer is *truly* included in the model's output. We adopt the textual entailment task (a.k.a. Natural Language Inference) for judgment, which has been used in Q&A evaluation [1] and hallucination detection [2,3]. We are also willing to explore *more advanced* hallucination detection methods to evaluate LLM unlearning in the future.
>
> ---
>
> ***References:***\
> [1] Accurate and nuanced open-qa evaluation through textual entailment, ACL 2024.\
> [2] Comparing Hallucination Detection Methods for Multilingual Generation, arXiv.2402.10496v2.\
> [3] A Survey on Hallucination in Large Language Models: Principles, Taxonomy, Challenges, and Open Questions, arXiv.2311.05232.\
> [4] SemEval-2017 Task 1: Semantic Textual Similarity - Multilingual and Cross-lingual Focused Evaluation, ACL 2017.\
> [5] Semantic Answer Similarity for Evaluating Question Answering Model, ACL 2021.\
> [6] Tofu: A task of fictitious unlearning for llms, COLM 2024.\
> [7] Negative Preference Optimization: From Catastrophic Collapse to Effective Unlearning, COLM 2024.

---

> ### Author Response · Authors · 2024-11-25
> **Looking forward to further feedback**
>
> Dear Reviewer qSfC,
>
> Sorry for bothering you, but the discussion period is coming to an end in two days. Could you please let us know if our responses have alleviated your concerns? If there are any further comments, we will do our best to respond.
>
> Best,
>
> The Authors

---

### Author Response · Authors · 2024-11-23
**Looking forward to further feedback**

Dear Reviewers,

Thank you again for your valuable comments and suggestions, which are really helpful for us. We have posted responses to the proposed concerns and included additional experiment results.

We totally understand that this is quite a busy period, so we deeply appreciate it if you could take some time to return further feedback on whether our responses solve your concerns. If there are any other comments, we will try our best to address them.

Best,

The Authors

---

### Meta-Review · Area_Chair_MR6T · 2024-12-18

**Metareview:**

The paper focuses on LLM unlearning, which aims to remove specific content—especially sensitive or copyrighted information—without necessitating complete retraining. The authors propose three additional metrics to improve the evaluation of unlearning performance: token diversity, sentence semantics, and factual correctness. They categorize unlearning methods into untargeted and targeted approaches, analyzing their limitations. For untargeted unlearning, they introduce a method based on maximizing entropy to avoid hallucinations, while for targeted unlearning, they propose an answer preservation (AP) loss to prevent excessive ignorance.

Evaluation is conducted on three different scenarios (fictitious unlearning, continual unlearning, and real-world unlearning), with significant performance improvements of the proposed method observed in all tasks. In addition to some new insights the paper proposed about LLM unlearning, the paper also has defined new evaluation metrics and proposed novel methods for both untargeted and targeted unlearning approaches. Although there have been concerns regarding consistency of presented techniques, plausibility of proposed evaluation metrics, and quite a few presentation issues, all reviewers lean towards acceptance after the rebuttal phase.

**Additional Comments On Reviewer Discussion:**

Although there have been concerns regarding consistency of presented techniques, plausibility of proposed evaluation metrics, and quite a few presentation issues, all reviewers lean towards acceptance after the rebuttal phase.

---

### Decision · Program_Chairs · 2025-01-22

Accept (Poster)